# Soliton walls paired by polar surface interactions in a ferroelectric nematic liquid crystal

Bijaya Basnet[1,2,5], Mojtaba Rajabi [1,3,5], Hao Wang[1,5], Priyanka Kumari[1,2], Kamal Thapa[1,3], Sanjoy Paul [1], Maxim O. Lavrentovich [4] & Oleg D. Lavrentovich [1,2,3✉]

Surface interactions are responsible for many properties of condensed matter, ranging from crystal faceting to the kinetics of phase transitions. Usually, these interactions are polar along the normal to the interface and apolar within the interface. Here we demonstrate that polar in-plane surface interactions of a ferroelectric nematic $N_F$ produce polar monodomains in micron-thin planar cells and stripes of an alternating electric polarization, separated by 180° domain walls, in thicker slabs. The surface polarity binds together pairs of these walls, yielding a total polarization rotation by 360°. The polar contribution to the total surface anchoring strength is on the order of 10%. The domain walls involve splay, bend, and twist of the polarization. The structure suggests that the splay elastic constant is larger than the bend modulus. The 360° pairs resemble domain walls in cosmology models with biased vacuums and ferromagnets in an external magnetic field.

[1] Advanced Materials and Liquid Crystal Institute, Kent State University, Kent, OH 44242, USA. [2] Materials Science Graduate Program, Kent State University, Kent, OH 44242, USA. [3] Department of Physics, Kent State University, Kent, OH 44242, USA. [4] Department of Physics and Astronomy, University of Tennessee, Knoxville, TN 37996, USA. [5] These authors contributed equally: Bijaya Basnet, Mojtaba Rajabi, Hao Wang. ✉email: olavrent@kent.edu

omains and domain walls (DWs) separating them are important concepts in many branches of physics, ranging from cosmology and high-energy science[1] to condensed matter[2–4]. When the system cools down from a symmetric ("isotropic") state, it might transition into an ordered state divided into domains. For example, domains in solid ferroic materials such as ferromagnets and ferroelectrics exhibit aligned magnetic moments or electric polarization[2–4]. Within each domain, the alignment is uniform, following some "easy direction" set by the crystal structure. These easy directions are nonpolar, thus opposite orientations of the polar order are of the same energy. The boundary between two uniform domains is a DW, within which the polar ordering either gradually disappears or realigns from one direction to another. By applying a magnetic or electric field, one can control the domains and DWs, which enables numerous applications of ferroics, ranging from computer memory to sensors and actuators[2–4].

Recent synthesis and evaluation[5–22] of new mesogens with large molecular dipoles led to a demonstration of a fluid ferroelectric nematic liquid crystal ($N_F$) with a uniaxial polar ordering of the molecules[13,14]. The ferroelectric nature of $N_F$ has been established by polarizing optical microscopy observations of domains with opposite orientations of the polarization density vector $\mathbf{P}$ and their response to a direct current (dc) electric field $\mathbf{E}$[13,14]. The surface orientation of $\mathbf{P}$ is set by buffed polymer layers at glass substrates that sandwich the liquid crystal[13,14]. This sensitivity to the field polarity and in-plane surface polarity makes $N_F$ clearly different from its dielectrically anisotropic but apolar paraelectric nematic counterpart N.

In this work, we demonstrate that the surface polarity of in-plane molecular interactions produces stable polar monodomains in micron-thin slabs of $N_F$ and polydomains in thicker samples. The polar contribution to the in-plane surface anchoring potential is on the order of 10%. The quasiperiodic polydomains feature paired domain walls (DWs) in which $\mathbf{P}$ realigns by 360°. The reorientation angle is twice as large as the one in 180° DWs of the Bloch and Néel types that are ubiquitous in solid ferromagnets and ferroelectrics[2,3] and in a paraelectric nematic N[23]. The polar bias of the "easy direction" of surface alignment explains the doubled amplitude of the 360° DWs and shapes them as coupled pairs of 180° static solitons. The width of DWs, on the order of 10 μm, is much larger than the molecular length scale, which suggests that the space charge produced by splay of the polarization within the walls is screened by ions and that the splay modulus $K_1$ in $N_F$ is significantly higher than the bend $K_3$ counterpart. The enhancement of $K_1$ is evidenced by the textures of conic-sections in $N_F$ films with a degenerate in-plane anchoring, in which the prevailing deformation is bend. Numerical analysis of the DW structure suggests that $K_1/K_3 > 4$ in the $N_F$ phase of the studied DIO material.

## Results
We explore a material abbreviated DIO[7], synthesized as described in the Supplementary Figs. 1–7. On cooling from the isotropic (I) phase, the phase sequence is I−174 °C − N−82 °C − SmZ$_A$−66 °C −$N_F$−34 °C−Crystal, where SmZ$_A$ is an antiferroelectric smectic with a partial splay[24], geometrically reminiscent of the splay N model proposed by Mertelj et al.[10] The sandwich-type cells are bounded by two glass plates with layers of polyimide PI-2555 buffed unidirectionally. The plates are assembled in a "parallel" fashion, with the two buffing directions $\mathbf{R}$ at the opposite plates being parallel to each other. We use Cartesian coordinates in which $\mathbf{R} = (0, -1, 0)$ is along the negative direction of the $y$-axis in the $xy$ plane of the sample. The electric field is applied along the $y$-axis.

**Planar alignment**. The N and SmZ$_A$ phases show a uniform alignment of the optical axis (director $\hat{\mathbf{n}}$) along the rubbing direction $\mathbf{R}$, Fig. 1a, b. In the absence of the electric field, depending on the thickness $d$ of the liquid crystal layer, $N_F$ forms either polydomain structures, when $d > 3$ μm, Fig. 1c, or polar monodomains in thin samples, $d = 1–2$ μm, Fig. 1d. At the bounding plates, $\mathbf{P}$ and $\hat{\mathbf{n}}$ are parallel to the surface, as evidenced by the measurement of optical retardance $\Gamma = 250$ nm at wavelength $\lambda = 535$ nm of a cell with $d = 1.35$ μm, which yields the DIO birefringence $\Delta n = \Gamma/d = 0.19$, close to the values reported by other groups[22,24]. Similar values of $\Delta n$ are obtained in homogeneous (free of DWs) regions of thicker cells, Supplementary Fig. 8. The monocrystal textures of thin cells and homogeneous regions of thick cells, Fig. 2a, become extinct when $\mathbf{P}$ and $\hat{\mathbf{n}}$ are parallel to the direction of polarizer or analyzer of a polarizing optical microscope (POM). These facts demonstrate planar alignment with little or no "pretilt" and exclude the possibility of director twist in DW-free regions of both thin and thick cells. The planar monocrystal structure of cells with parallel assembly of unidirectionally buffed substrates should be contrasted to the textures in cells with antiparallel assembly, in which $\mathbf{P}$ and $\hat{\mathbf{n}}$ twist along the normal $z$-axis[13,14].

The planar alignment avoids a strong surface charge. Even a small tilt $\psi \sim 5°$ of $\mathbf{P}$ from the $xy$ plane would produce a surface charge density $P_z \sim P\psi \sim 4 \times 10^{-3}$ C m$^{-2}$, which is larger than the typical surface charge $(10^{-4} - 10^{-5})$C m$^{-2}$ of adsorbed ions reported for nematics[25,26]; here $P \approx 4.4 \times 10^{-2}$ C m$^{-2}$ is the polarization of DIO[7]. Therefore, we expect that the out-of-plane (zenithal) polar anchoring is much stronger than the in-plane azimuthal anchoring.

**Ferroelectric monodomains in thin $N_F$ cells**. Thin cells, $d = 1 - 2$ μm, filled in the N phase at 120 °C, and cooled down with the rate 2 °C/min, show a monodomain texture, with the polarization $\mathbf{P} = P(0, 1, 0)$ antiparallel to $\mathbf{R} = (0, -1, 0)$, Fig. 1d. A dc electric field $\mathbf{E} = E(0, 1, 0)$ directed along $\mathbf{P}$ and of an amplitude $E = (1 - 10)$ kV/m causes no textural changes, while the opposite field polarity reorients $\hat{\mathbf{n}}$ and $\mathbf{P}$ beginning with $E_\downarrow = -1.0$ kV/m, Fig. 1d. As the field increases, the optical retardance $\Gamma$ diminishes, Fig. 1d, which indicates that $\hat{\mathbf{n}}$ twists away from the rubbing direction in the bulk. Above a critical field $E_c = -11$ kV/m, the surface anchoring that keeps $\mathbf{P}$ antiparallel to $\mathbf{R}$ ($\mathbf{P}\uparrow\downarrow\mathbf{R}$) is broken, and a uniformly aligned state $\mathbf{P}\downarrow\downarrow\mathbf{R}$ nucleates and propagates across the cell, swiping away the twisted state. Once formed, the $\mathbf{P}\downarrow\downarrow\mathbf{R}$ state is stable for days, even in the absence of the field. A field $\mathbf{E} = E(0, 1, 0)$ that is antiparallel to $\mathbf{R}$ realigns $\mathbf{P}$ back to the ground state $\mathbf{P}\uparrow\downarrow\mathbf{R}$, beginning with $E_\uparrow = 0.6$ kV/m, which is noticeably lower than $|E_\downarrow|$, Fig. 1e. Figure 1f schematizes the polarization realignment from the local anchoring minimum $\mathbf{P}\downarrow\downarrow\mathbf{R}$ to the global one at $\mathbf{P}\uparrow\downarrow\mathbf{R}$, which is accompanied by the formation of horizontal left- and right-twisted 180° DWs of the Bloch type near the plates. Multiple cycles of switching leave $E_\uparrow$ and $E_\downarrow$ intact, which means that the electric field realigns the polarization $\mathbf{P}$ in the liquid crystal bulk but does not switch the polarity of the rubbing direction $\mathbf{R}$. Note also that heating the material into I and then cooling it down to $N_F$ restores $\mathbf{P}$ antiparallel to $\mathbf{R}$.

**Polar character of in-plane anchoring of planar $N_F$ cells**. The difference in the electric fields $|E_\downarrow|$ and $|E_\uparrow|$ that deviate $\mathbf{P}$ from the states $\mathbf{P}\uparrow\downarrow\mathbf{R}$ and $\mathbf{P}\downarrow\downarrow\mathbf{R}$, respectively, demonstrates that the in-plane anchoring in the cells with the parallel assembly of the buffed plates exhibits two energy minima, one global at $\varphi = 0$, and another local at $\varphi = \pm\pi$. Here, $\varphi$ is the angle that $\mathbf{P}$ makes

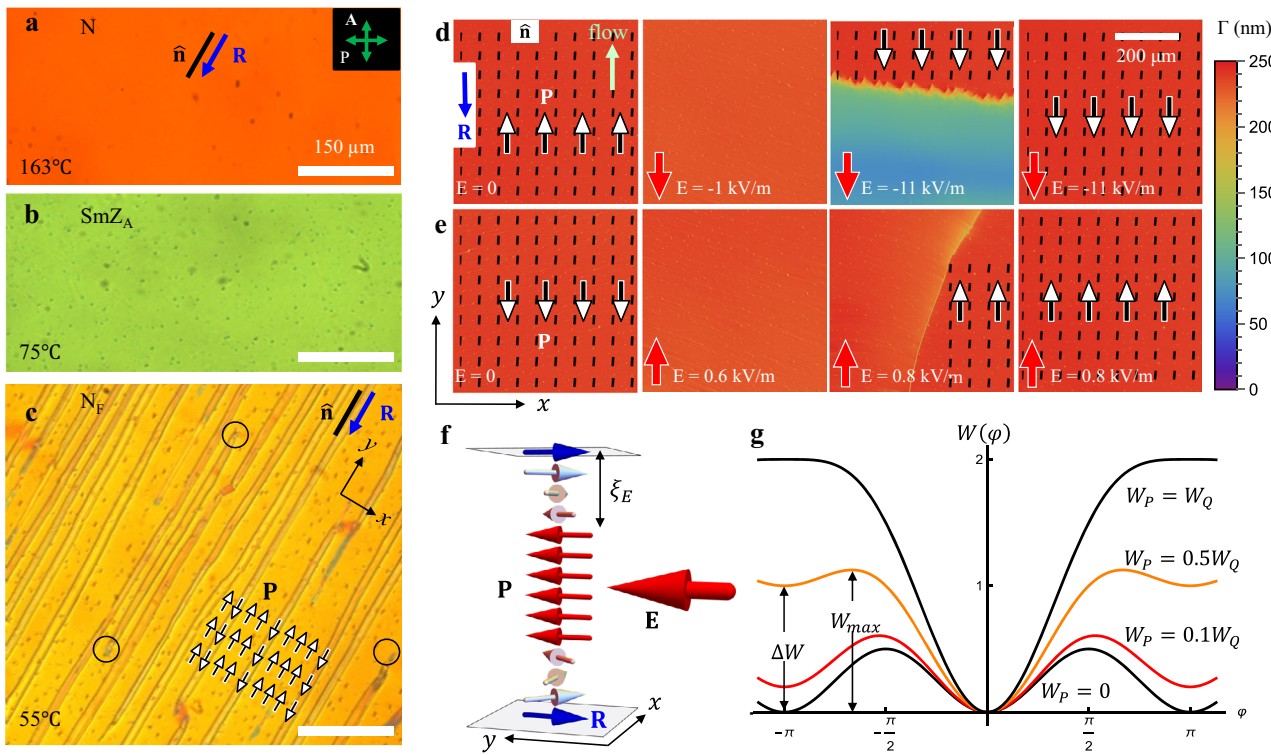

**Fig. 1 DIO textures in planar cells with parallel assembly. a–c** Polarizing optical microscopy of a thick $d = 4.7\,\mu m$ sample and **d, e** PolScope Microimager textures of a thin 1.35 μm sample. **a, b** Uniform N and SmZ$_A$ textures, respectively. **c** polydomain N$_F$ texture; the polarization **P** is antiparallel to the rubbing direction **R** in the wider domains and is parallel to **R** in the narrower domains; two 180° DWs enclosing the narrow domain reconnect (circles mark some reconnection points). **d** Field-induced realignment of **P** from the direction −**R** to **R**. **e** Reversed field polarity realigns **P** back into the ground state **P** ↑↓ **R**. **f** Scheme of **P** reorientation in (**e**); there are two 180° twist DWs of the Bloch type near the plates. **g** Azimuthal surface anchoring potential for different ratios of the polar $W_P$ and apolar $W_Q$ coefficients.

with the $y$-axis. The azimuthal surface anchoring potential that captures these features is

$$W(\varphi) = \frac{W_Q}{2}\sin^2\varphi - W_P(\cos\varphi - 1) \qquad (1)$$

where $W_Q \geq 0$ and $W_P \geq 0$ are the apolar (quadrupolar, or nematic-like) and polar anchoring coefficients, respectively, Fig. 1g. This form follows the one proposed by Chen et al.[14] and places a global minimum at $\varphi = 0$. When $W_P = 0$, the anchoring is polarity-insensitive, and the minima at $\varphi = 0, \pm\pi$ are of an equal depth. As $W_P$ increases, the minima at $\varphi = \pm\pi$ raise to the level $\Delta W = 2W_P$ and become local, until disappearing at $W_P \geq W_Q$, Fig. 1g. The energy barrier $W_{max} = W_Q(1 + \omega)^2/2$ at $\varphi = \arccos(-\omega)$ separates the global and local minima; $\omega = W_P/W_Q$ is the relative strength of the in-plane polar anchoring.

The surface anchoring torques[27] $\frac{\partial W(\varphi)}{\partial\varphi}\big|_{z=0,d} = (W_Q\sin\varphi\cos\varphi + W_P\sin\varphi)\big|_{z=0,d}$ resist the realigning action of the field, Fig. 1f. For a small deviation from the preferred state $\varphi = 0$, the torque is $W_Q + W_P$; for a deviation from the metastable state $\varphi = \pm\pi$ the torque is weaker, $W_Q - W_P$. These torques compete with the elastic torque $K_2/\xi_E = \sqrt{K_2PE}$ caused by the field-induced twist of **P** in subsurface regions of a characteristic thickness $\xi_E = \sqrt{K_2/PE}$, where $K_2$ is the twist elastic constant, Fig. 1f. The difference in the surface torques explains the difference in the reorienting fields, $\frac{W_Q + W_P}{W_Q - W_P} = \sqrt{\left|\frac{E_\downarrow}{E_\uparrow}\right|} \approx 1.3$, which allows one to determine the relative strength of the polar anchoring, $\omega = W_P/W_Q \approx 0.13$. The measured $E_\uparrow = 0.6\,\text{kV/m}$, $E_\downarrow =$

$-1.0\,\text{kV/m}$, reported[7] $P = 4.4 \times 10^{-2}\,\text{C/m}^2$, and a reasonable assumption[27] $K_2 \approx 5\,\text{pN}$, lead to the estimates $\xi_E \approx 0.3\,\mu m$, $W_Q \approx 1.3 \times 10^{-5}\,\text{J/m}^2$, and $W_P \approx 1.7 \times 10^{-6}\,\text{J/m}^2$. The estimated $W_Q$ is within the range reported for nematics at rubbed polyimides[28,29].

Note here that in the thin cell under study, the material was filled in the N phase at 120 °C by a capillary flow along the −**R** direction, Fig. 1d. Filling a cell by a flow at 120 °C along **R** yields $E_\downarrow = -1.4\,\text{kV/m}$ and $E_\uparrow = 1.0\,\text{kV/m}$, which implies a weaker polar bias: $\omega \approx 0.08$. This flow effect on the surface anchoring deserves further study, but to describe the polydomain patterns in thick cells, we avoid it by filling the cells in at 180 °C and then rapidly cooling the sample through the N phase with a rate 30 °C/min, followed by slow cooling through SmZ$_A$ and N$_F$ with the rate 2 °C/min. Thin $d = 1.1\,\mu m$ monodomain samples show $E_\downarrow = -0.4\,\text{kV/m}$, $E_\uparrow = 0.3\,\text{kV/m}$, which yields $\omega \approx 0.07$. With the values of $P$ and $K_2$ above, one estimates $W_Q \approx 8.8 \times 10^{-6}\,\text{J/m}^2$, and $W_P \approx 0.63 \times 10^{-6}\,\text{J/m}^2$. In what follows, we discuss the data for cells filled in the isotropic phase at 180 °C; the domain structures are similar to those in cells filled at 120 °C.

**Ferroelectric domains in thick planar N$_F$ cells.** Cooling cells of thickness $d = 3 - 16\,\mu m$ from the SmZ$_A$ phase results in a quasiperiodic domain texture of N$_F$, Fig. 1c, with alternating homogeneous **P** ↑↓ **R** and **P** ↓↓ **R** stripes, as established by the response to the in-plane electric field, Figs. 2 and 3. For example, the cell of thickness $d = 4.7\,\mu m$ shows relatively wide (5–150 μm)

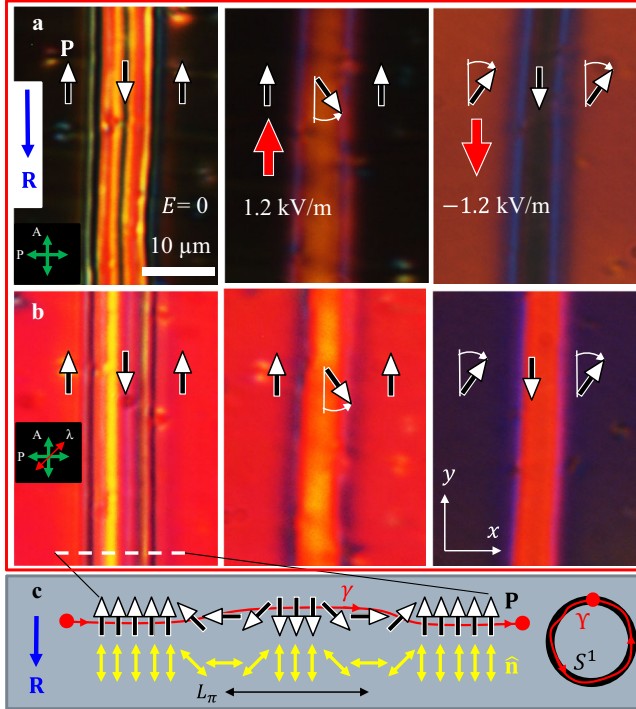

**Fig. 2 Topologically stable 360° W-pairs of DWs. a** Textures observed between crossed polarizers with **P** ↓↓ **R** in the narrow central domain separated by two bright 180° DWs from the wide domains with **P** ↑↓ **R** at the periphery; the electric field realigns **P** in the narrow or wide domains, depending on the field polarity. **b** The same textures, observed with an optical compensator that allows one to establish the reorientation direction of **P**. **c** Topologically nontrivial structure of the 360° W-pair of DWs; along the line $\gamma$, the polarization vector **P** rotates by 360°, thus covering the order parameter space $S^1$ once, which yields the topological charge $Q = 1$. Cell thickness $d = 4.7\,\mu m$.

regions in which **P** ↑↓ **R** and narrow (1–2 $\mu m$) regions in which **P** ↓↓ **R**, respectively, Figs. 1c, 2 and 3. Once formed, the domains remain stable for days. Repeating heating-cooling cycles, even following a crystallization or transition into the isotropic phase, reproduces the same qualitative $N_F$ patterns.

Both narrow and wide domains are extinct when aligned along the polarizers of POM, Fig. 2a, and show optical retardance $\Gamma = 900$ nm at $\lambda = 535$ nm and $d = 4.7$ nm, which means that $\Gamma/d$ coincides with $\Delta n$ and thus **P** and $\hat{\mathbf{n}}$ must be in the $xy$ plane of the cell.

**Paired domain walls of W and S types in thick planar $N_F$ cells.** Domains of opposite polarization are separated by DWs. Within each DW, **P** and $\hat{\mathbf{n}}$ must realign by 180°. The DWs enclosing the narrow domains always exist and terminate in pairs, Figs. 1c, 2 and 3, so that the reorientation within the DW pair is 360° in the plane of the sample. To elucidate the structures in a greater detail, we use thicker cells ($d = 6.8\,\mu m$), in which the narrow domains are slightly wider, Supplementary Fig. 9, and perform POM observations with monochromatic light, using a blue interferometric filter of a central wavelength $\lambda = 488$ nm, full width at half maximum (FWHM) 1 nm, and a red filter ($\lambda = 632.8$ nm, FWHM 1 nm), Fig. 4.

In crossed polarizers aligned parallel and orthogonal to the DWs, the regions in which **P** ↑↓ **R** ($\varphi = 0, 2\pi$) and **P** ↓↓ **R** ($\varphi = \pi$) appear dark in both polychromatic, Fig. 4a, and blue light, Fig. 4b, c. The blue filter observations reveal that the regions located approximately

half-way between $\varphi = 0, 2\pi$ and $\varphi = \pi$ are also dark, apparently corresponding to $\varphi = \pi/2, 3\pi/2$, Fig. 4b, c. The dark stripes associated with $\varphi = 0, \pi/2, \pi, 3\pi/2, 2\pi$, are separated by bright stripes, corresponding to intermediate $\varphi$'s, Fig. 4a–c. The textures in Fig. 4a–c make it clear that the described DWs are indeed walls with a 360° reorientation of **P** and $\hat{\mathbf{n}}$, as opposed to the "bend texture with line disclination" of other $N_F$ materials presented by Li et al.[21], 180° surface disclination lines and 180° DWs described by Chen et al.[13] and Li et al.[21]. The transmitted intensity profile in Fig. 4c allows one to introduce the characteristic width parameters of the DW pairs: distances $L_{\pi/2}$ between the two central bright stripes, $L_\pi$ between two dark narrow stripes, and $L_{3\pi/2}$ between two outermost stripes. These distances, although small (8–15 $\mu m$), are clearly wider than the cores of singular disclinations, and 180° walls or surface disclinations described previously. Importantly, besides the in-plane 360° reorientation of **P** and $\hat{\mathbf{n}}$, the textures in $d = 6.8\,\mu m$ cells also suggest tilts of these vectors away from the cell's $xy$ plane, as described below.

When the crossed polarizers are at 45° with respect to the DWs, polychromatic light observations reveal the same interference colors in narrow ($\varphi = \pi$) and wide ($\varphi = 0, 2\pi$) domains, Fig. 4d. The chosen $d = 6.8\,\mu m$ allows us to achieve destructive interference of the ordinary and extraordinary waves in POM observations with a red filter ($\lambda = 632.8$ nm), at which $\Delta n = 0.189$, since the factor $\frac{\pi d \Delta n}{2\lambda} = 3.19$ associated with the interference of the two modes[27] is close to $\pi$. Although the crossed polarizers are at 45° to the DWs, destructive interference causes extinction in the regions with $\varphi = 0, \pi$, and $2\pi$, where **P** and $\hat{\mathbf{n}}$ are in the $xy$ plane; regions $\varphi = \pi/2, 3\pi/2$ also appear dark. A notable exception are four narrow peaks of transmission, at $0 < \varphi < \pi/2$, $\pi/2 < \varphi < \pi$, $\pi < \varphi < 3\pi/2$, and $3\pi/2 < \varphi < 2\pi$, Fig. 4f, which signal the appearance of a polar $z$-component of **P** and $\hat{\mathbf{n}}$.

There are two types of the 360° DW pairs. In the first, called W-pairs because of the shape of the director field, Fig. 2c, **P** rotates by 180° in the same fashion in both DWs, either clockwise (CW) or counterclockwise (CCW). In the second type, called 360° S-pairs for their geometry, Fig. 3f, the rotation directions alternate: if **P** rotates CW by 180° in one DW, it rotates CCW by 180° in the next one. The splay-bend schemes of Figs. 2c and 3f demonstrate only the topological features of the in-plane realignments; polar tilts and associated twists add to the complexity of the splay-bend and will be treated in the section on numerical simulations.

The difference between the W- and S-pairs is topological, as illustrated by mappings of the oriented line $\gamma$ threaded through the DWs pair and the enclosed domain, into the order parameter space, a circle $S^1$ [27], Figs. 2c and 3f. Each point on $S^1$ corresponds to a certain $\varphi$. The line $\gamma$ in Fig. 2c produces a CCW-oriented closed contour $\Upsilon$ encircling $S^1$ once. The W-pair of CCW walls thus carries a topological charge $Q = 1$ [27]. A DW pair with a CW 360° rotation of **P** would carry $Q = -1$. Neither could be transformed into a uniform state $Q = 0$ without breaking the surface anchoring and overcoming a large elastic energy barrier. S-pairs of 180°-walls with alternating sense of rotations are topologically trivial, $Q = 0$: the corresponding contour $\Upsilon$ does not encircle $S^1$ fully and could be contracted into a single point $\varphi = 0$ without the need to overcome the elastic energy barrier, Fig. 3g.

**Width of domain walls $N_F$ cells and electrostatic effects.** The elastic energy density stored within a DW, $\frac{K}{L_\pi^2} \sim \frac{k_B T}{a L_\pi^2}$, where $K$ is the average Frank elastic constant, $k_B T$ is Boltzmann's energy, $a \sim 1$ nm is the molecular size, and $L_\pi \approx (5-20)\,\mu m$ is the characteristic width of a DW pair, defined as the distance between the $x$-coordinates of two bend regions, $\varphi = \pi/2$ and $3\pi/2$, Figs. 2c

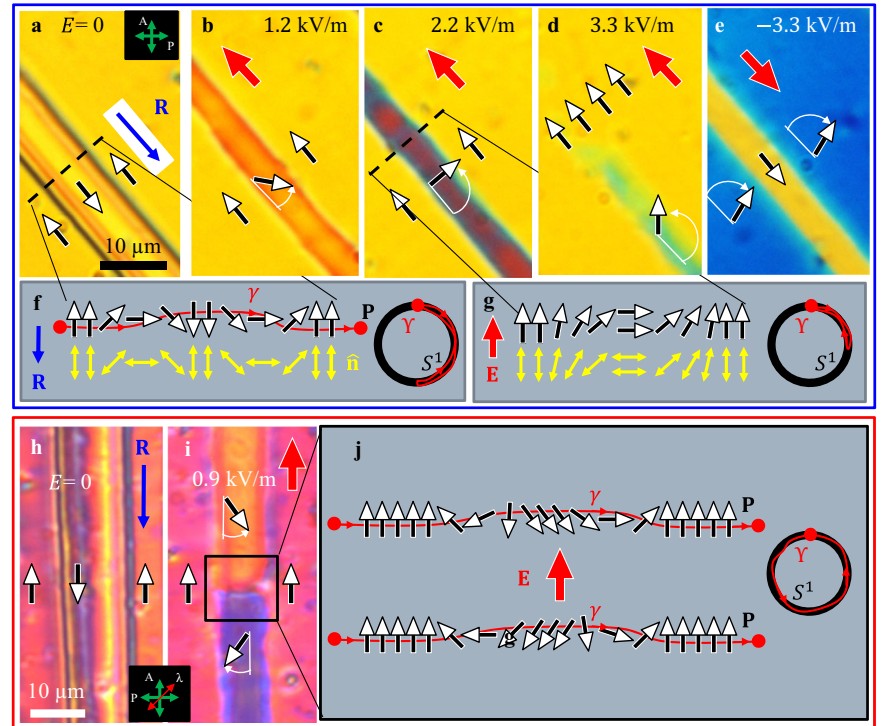

**Fig. 3 Electric field switching of 360° DW pairs. a–d** POM textures of topologically trivial S-pair that are smoothly realigned into a uniform state by the electric field of an appropriate polarity. **e** An opposite field polarity tilts **P** in two wide domains, but does not cause a complete reorientation, contrary to the case of the narrow domain in (**d**). **f** Topological scheme of the S-pair shown in (**a**). **P** rotates CW in the left DW and CCW in the right DW, thus $Q = 0$. **g** Topological scheme of the S-pair shown in (**c**). **P** in the central narrow domain could rotate only CCW as the field increases. **h, i** POM textures (with an added waveplate) of a topologically stable 360° W-pair of DWs, $Q = 1$; increase of the electric field could cause both CW and CCW rotations of **P** within the same DW pair, as schematized in (**j**). Cell thickness 4.7 μm in all textures.

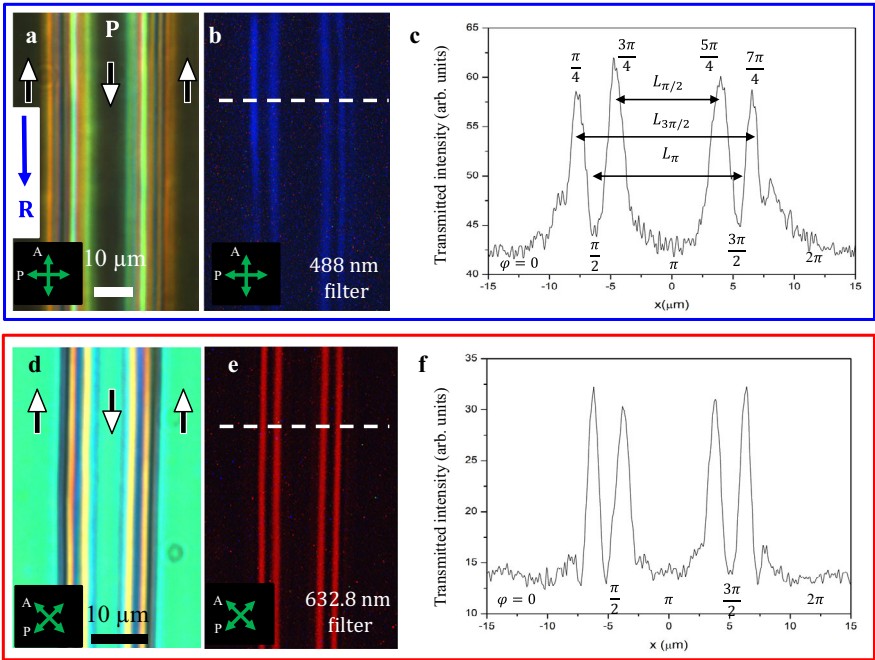

**Fig. 4 Fine structure of 360° DW pairs. a** Polychromatic texture of a DW pair running parallel to one of the crossed polarizers; wide **P** ↑↓ **R** ($\varphi = 0, 2\pi$) and narrow **P** ↓↓ **R** ($\varphi = \pi$) domains are extinct. **b** The same texture, observed with a blue filter; the stripes with $\varphi = \pi/2$ and $3\pi/2$ where **P** is perpendicular to the DWs are also extinct. **c** Transmitted light intensity along the dashed line in (**b**). **d** Polychromatic texture of a DW pair running at 45° to the crossed polarizers; wide **P** ↑↓ **R** ($\varphi = 0, 2\pi$) and narrow **P** ↓↓ **R** ($\varphi = \pi$) domains show similar optical retardance. **e** The same texture, observed with a red filter that yields destructive interference at locations $\varphi = 0, \pi/2, \pi, 3\pi/2, 2\pi$. **f** Transmitted light intensity along the dashed line in (**e**). Cell thickness 6.8 μm in all textures.

**Fig. 5 Polarizing microscopy textures of DIO at the glycerin substrate. a** N film shows $2\pi$ domain splay-bend walls. **b, c** $N_F$ texture of conic-sections with prevailing circular bend; in (**b**), elliptical defects separate regions between mostly circular bend and mostly uniform **P** field, while in (**c**), hyperbolic shapes separate domains with predominantly circular bend. Film thickness 7 μm in (**a, c**), and 5 μm in (**b**); **n̂** is depicted by white lines.

and 4c, f, is much lower than the energy density $\frac{k_B T}{a^3}$ of the orientational order. Therefore, $\mathbf{P} \parallel \hat{\mathbf{n}}$ and realignment of **P** preserves the magnitude $P$. This feature makes the observed DWs similar to Néel DWs in ferroics, as opposed to Ising DWs, in which $P \rightarrow 0$.

Reorientation of **P** within each DW generates a "bound" space charge of density $\rho_b = -\text{div } \mathbf{P}$. If the polarization charge is not screened by ionic charges, then the balance of the elastic energy (per unit area of the wall) $\frac{K}{L_\pi}$ and the electrostatic energy $\frac{P^2 L_\pi}{\varepsilon \varepsilon_0}$ suggests[30] that a DW would be of a nanoscale width, equal the so-called polarization penetration length $\xi_P = \sqrt{\frac{\varepsilon \varepsilon_0 K}{P^2}}$, where $\varepsilon_0$ is the electric constant, $\varepsilon$ is the dielectric permittivity of the material. For the DIO polarization density[7] $P = 4.4 \times 10^{-2} \text{C/m}^2$ and assumed $K = 10$ pN, $\varepsilon = 10$, one finds $\xi_P \approx 1$ nm, much smaller than the observed $L_\pi$, Fig. 4. Note here that the estimated $\varepsilon$ is lower than the often reported value $10^4$, which might be exaggerated by the effect of polarization realignment[31]. The polarization charge of density $\rho_b \sim \frac{P}{L_\pi} \sim (0.2 - 0.9) \times 10^4 \text{C/m}^3$ at the splay region of a DW should be screened by mobile free charges, supplied by ionic impurities, ionization, and absorption effects. To achieve a comparable screening charge $\rho_f \sim en \sim (0.2 - 0.9) \times 10^4 \text{C/m}^3$, where $e = 1.6 \times 10^{-19} C$ is the elementary charge, the concentration of ions at the DW should be $n \sim \left(10^{22} - 10^{23}\right) / \text{m}^3$. A high ion concentration $n \sim 10^{23} / \text{m}^3$ has been reported as a volume-averaged value for ferroelectric smectics[32], although conventional nematics usually yield smaller values[33], $n \sim \left(10^{20} - 10^{22}\right) / \text{m}^3$. It is reasonable to assume that even when the volume-averaged $n$ is less than $n \sim \left(10^{22} - 10^{23}\right) / \text{m}^3$, mobile charges could move from the uniform regions of the material and accumulate at local concentrations sufficient to screen the splay-induced polarization charge.

As envisioned by Meyer[34] and detailed theoretically in the subsequent studies[35–37], the ionic screening enhances the splay elastic constant $K_1$ associated with $(\text{div } \hat{\mathbf{n}})^2$ in the Frank–Oseen free energy density: $K_1 = K_{1,0}\left(1 + \lambda_D^2/\xi_P^2\right)$, where $K_{1,0}$ is the bare modulus, of the same order as the one normally measured in a conventional paraelectric N, and $\lambda_D = \sqrt{\frac{\varepsilon \varepsilon_0 k_B T}{ne^2}}$ is the Debye screening length, which, for the typical parameters specified above and $n = 10^{23} / \text{m}^3$, is on the order of 10 nm. With $\lambda_D \sim 10$ nm, $\xi_P \sim 1$ nm, the enhancement factor, $\frac{\lambda_D^2}{\xi_P^2} \sim 10^2$, could be strong. Thus, $K_1$ in $N_F$ can be much larger than $K_1$ in N. Very

little is known about the elastic constants in the N phase of ferroelectric materials and practically nothing is known about the elasticity of $N_F$. Chen et al.[24] measured $K_1 \approx 10 K_2$ in the N phase of DIO and expected[37] $K_1 \approx 2$ pN. Mertelj et al.[10] reported that in the N phase of another ferroelectric material RM734, $K_1$ is even lower, about 0.4 pN. Since the bend elastic constant $K_3$ of $N_F$ is not supposed to experience an electrostatic renormalization, it is expected to be a few tens of pN; for example, Mertelj et al.[10] found $K_3 \approx 10{-}20$ pN for the N phase of RM734. Therefore, the ratio $\kappa = K_1/K_3$ in $N_F$ could be larger than 1, ranging from a single-digits value to $\sim 10^2$. The next section presents qualitative evidence that $K_1 > K_3$ in $N_F$.

**Prevalence of bend in $N_F$ films with degenerate in-plane anchoring.** The textures of N and $N_F$ are strikingly different when there is no in-plane anchoring. Figure 5 shows the textures of thin ($d = 5 - 7$ μm) films of DIO spread onto glycerin; the upper surface is free. Thermotropic N films are known to form $2\pi$ domain walls of the W type, stabilized by the hybrid zenithal anchoring, tangential at the glycerin substrate and tilted or homeotropic at the free surface[38]; these $2\pi$ domain walls contain both splay and bend and are clearly distinguished in DIO as bands with four extinction bands, Fig. 5a. The $N_F$ textures feature an optical retardance that is consistent with the director being tangential to the film. The most important feature is that the curvature lines of **P** and $\hat{\mathbf{n}}$ are close to circles and circular arches, Fig. 5b, c, which implies prevalence of bend and signals that splay is energetically costly. One often observe disclinations of strength $+1$ with predominant bend, Fig. 5b, c. The regions with $+1$ disclinations are separated from regions with a straight or nearly straight **P** by defects shaped as parts of ellipses and parabolas, Fig. 5b, while two neighboring domains with a $+1$ disclination in each are separated by hyperbolic defects, Fig. 5c.

The conic-sections textures (CSTs) of $N_F$ in Fig. 5b, c resemble focal conic domain (FCD) textures of a smectics A, in which the layers are shaped as the so-called Dupin cyclides[27] that preserve equidistance and avoid bend and twist of the normal to the layers (which is the smectic director). The distinct feature of the Dupin cyclides is that their focal surfaces reduce to conic-sections, such as a confocal ellipse and hyperbola, or pairs of parabolas. The CSTs in Fig. 5b, c shows similar conic-sections as the boundaries between regions of different director curvatures. In $N_F$, the director avoids splay; twist is not prohibited, but the degenerate anchoring does not require it. The FCD textures in a smectic A reflect the inequality $K_3 \gg K_1$, while the CSTs in $N_F$ suggest $K_1 > K_3$; a detailed analysis of CSTs will be presented elsewhere. In what follows, we explore the DW pairs in planar samples theoretically, first in a simplified one-constant approximation,

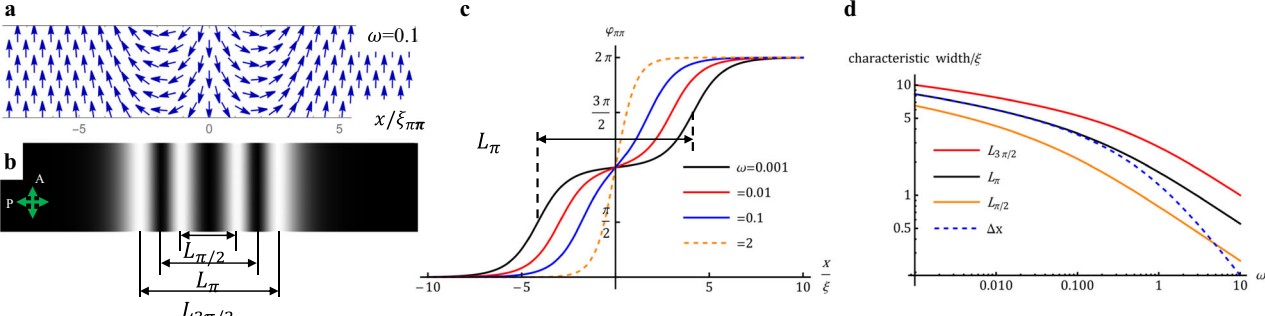

**Fig. 6 Equilibrium $\pi\pi$ soliton-soliton pairs described by Eq. (7). a** In-plane polarization field for $\omega = 0.1$. **b** The corresponding texture observed between crossed polarizers with the intensity of transmitted light calculated with Eq.(8). **c** Polarization profile $\varphi_{\pi\pi}(x)$ for different surface anchoring anisotropies $\omega$; the separation $L_\pi$ between two extinction bands at $\varphi_{\pi\pi} = \pi/2$ and $\varphi_{\pi\pi} = 3\pi/2$ is shown for the profile with $\omega = 0.001$. **d** Characteristic widths of the $\pi\pi$ soliton-soliton pairs defined in (**b**) vs. $\omega$; note that $\Delta x \cong L_\pi$ for $\omega < 0.1$, but $\Delta x < L_\pi$ for $\omega > 0.1$.

and then accounting for the possibility of elastic anisotropy $K_1 > K_3$ and non-planar geometry of the director.

**Balance of elasticity and surface anchoring in N$_F$ domain walls.** The observed coexistence of the wide $\mathbf{P}\uparrow\downarrow\mathbf{R}$ and narrow $\mathbf{P}\downarrow\downarrow\mathbf{R}$ domains in planar cells results from the two-minima surface potential $W(\varphi)$, Fig. 1g, balanced by the bulk elasticity of N$_F$. According to the experiments, the director within the DW pair experiences a reorientation by $2\pi$ along the $x$ axis, which must incorporate both splay and bend, Figs. 2–4. The experimental data in Fig. 4e, f also demonstrate a polar tilt towards the $z$-axis; this tilt adds a twist of $\mathbf{P}$. To make the theoretical analysis tractable, the overall director field could be approximated as

$$\hat{\mathbf{n}} = \left[\sin\varphi(x)\cos\theta(x,z), \cos\varphi(x)\cos\theta(x,z), \sin\theta(x,z)\right], \quad (2)$$

where the azimuthal angle $\varphi(x)$ between $\mathbf{P}$ and the $y$-axis varies only along the $x$-axis and the polar angle $\theta(x,z)$ between $\mathbf{P}$ and the $xy$ plane could change along both the $x$- and $z$-axes. Far from the DW pair, the boundary conditions are $\varphi(x) = \theta(x,z) = 0$. We also measure $\theta(x,z) = 0$ at the locations with $\varphi = 0$, $\pi$, and $2\pi$, where optical retardance is close to $d\Delta n$, Fig. 4e, f. Since the polar tilt at the bounding plates is penalized by a large surface charge, we assume that the zenithal polar anchoring is infinitely strong and approximate the bulk variations of the polar angle as

$$\theta(x,z) = \theta_a(x)\sin\frac{2\pi z}{d}, \quad (3)$$

which satisfies the boundary condition $\theta(x,z) = 0$ at $z = \pm d/2$; $\theta_a$ is the tilt amplitude.

The Frank–Oseen free energy with the bulk, saddle-splay, and the azimuthal surface anchoring terms reads

$$F = F_b + F_{24} + W = \frac{1}{2}\int dV \left[K_1(\text{div}\hat{\mathbf{n}})^2 + K_2(\hat{\mathbf{n}}\cdot\text{curl}\hat{\mathbf{n}})^2\right.$$
$$\left. + K_3(\hat{\mathbf{n}}\times\text{curl}\hat{\mathbf{n}})^2 - 2K_{24}\text{div}(\hat{\mathbf{n}}\text{div}\hat{\mathbf{n}} + \hat{\mathbf{n}}\times\text{curl}\hat{\mathbf{n}})\right] \quad (4)$$
$$+ \int dxdy\left[W_Q\sin^2\varphi - 2W_P(\cos\varphi - 1)\right],$$

where $K_1$, $K_2$, $K_3$, and $K_{24}$ are the elastic constants of splay, twist, bend, and saddle-splay, respectively. The equilibrium director field $\hat{\mathbf{n}}||\mathbf{P}$ minimizing the free energy in Eq. (4) could be found only numerically. However, analytical solutions useful for the understanding of the DW pairs could be found if $\theta(x,z) = 0$ and $K_1 = K_3 = K$; the planar geometry with $\theta = 0$ excludes twists.

**Analytical solutions for planar domain walls.** Setting the variation of the energy (4) to zero leads to the first integral of the

Euler–Lagrange equation:

$$\frac{Kd}{2W_Q}\left(\frac{\partial\varphi}{\partial x}\right)^2 - \sin^2\varphi + 2\omega(\cos\varphi - 1) = \text{const.} \quad (5)$$

For an apolar anchoring, $\omega = 0$, and the boundary conditions $\frac{\partial\varphi}{\partial x}(\pm\infty) = 0$, $\varphi(-\infty) = 0$, $\varphi(\infty) = \pi$, the constant of integration is 0 and the solution

$$\varphi_\pi(x) = 2\arctan e^{\frac{x}{\xi}} \quad (6)$$

represents a static $\pi$-soliton with a characteristic width $\xi = \sqrt{\frac{Kd}{2W_Q}}$, within which $\mathbf{P}$ realigns into $-\mathbf{P}$. This solution is an "inversion wall" of the Néel type observed by Nehring and Saupe in planar N cells[23]. The energy per unit length of each $\pi$-soliton, obtained by integrating $f$ with $W_P=0$ over the range $-\infty < x < \infty$, is finite, $F_\pi = 2\sqrt{2KdW_Q}$.

When $W_P > 0$, the single-wall solution (6) is no longer valid since $\varphi = \pm\pi$ is only a local minimum of $W(\varphi)$. With $W_P > 0$, Eq. (5) is a double-sine-Gordon equation, extensively studied in high-energy physics and cosmology[39] and physics of ferromagnets[4], in which case the analogs of the surface $W_Q$ and $W_P$ terms are of a bulk nature, associated, e.g., with the crystal anisotropy of a ferromagnet and the external magnetic field, respectively. With boundary conditions $\frac{\partial\varphi}{\partial x}(\pm\infty) = 0$, $\varphi(\pm\infty) = 0, 2\pi$, among the solutions of Eq. (4) are topologically protected $\pi\pi$ soliton-soliton pairs with 360° in-plane reorientation of $\mathbf{P}$ and a topological charge $Q = \pm 1$:

$$\varphi_{\pi\pi}(x) = \pm 2\arctan\left[\exp\left(\frac{x}{\xi_{\pi\pi}} + \frac{\delta_{\pi\pi}}{2}\right)\right]$$
$$\pm 2\arctan\left[\exp\left(\frac{x}{\xi_{\pi\pi}} - \frac{\delta_{\pi\pi}}{2}\right)\right] \quad (7)$$

where $\xi_{\pi\pi} = \xi\sqrt{\frac{1}{1+\omega}}$, $\delta_{\pi\pi} = 2\text{arcsinh}\sqrt{\frac{1}{\omega}}$; "+" signs correspond to a $Q = 1$ pair in Fig. 2c, e and Supplementary Fig. 10a. The solution is a superposition of two $\pi$-walls located at $x = \pm\frac{\xi_{\pi\pi}\delta_{\pi\pi}}{2}$ and limiting a stripe of a nearly uniform $\mathbf{P}\downarrow\downarrow\mathbf{R}$, Fig. 6a. The $\pi\pi$-soliton (7) is topologically equivalent to the 360° DW pair of the W type in Figs. 2, 3h, 3i and 4. The energy per unit length of this $\pi\pi$-soliton is finite: $F_{\pi\pi} = 2F_\pi[\sqrt{1+\omega} + \omega\text{arccoth}\sqrt{1+\omega}]$.

The intensity of unpolarized monochromatic light, transmitted through two crossed polarizers enclosing a birefringent sample with a DW pair described by Eq. (7) and running parallel to one of the polarizers[27],

$$I \propto \sin^2(2\varphi_{\pi\pi})\sin^2\left(\frac{\pi d\Delta n}{2\lambda}\right), \quad (8)$$

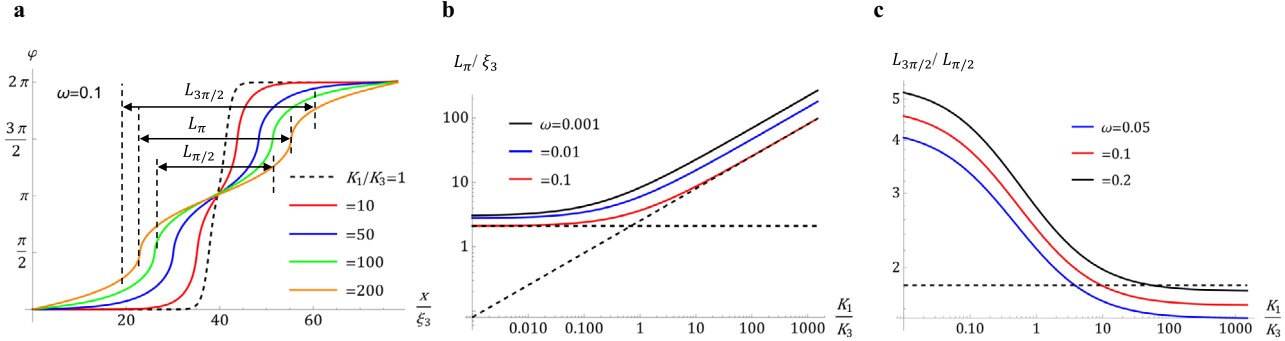

**Fig. 7 Equilibrium planar $\pi\pi$ soliton-soliton pairs for different splay and bend constants. a** Director profiles of DWs pairs for $\omega = 0.1$ and different elastic ratios $K_1/K_3$. **b** The width parameter $L_\pi$ vs $K_1/K_3$ for different anchoring anisotropies $\omega$. **c** Ratio of width parameters $L_{3\pi/2}/L_{\pi/2}$ vs $K_1/K_3$ for different anchoring anisotropies $\omega$; the dashed line shows $L_{3\pi/2}/L_{\pi/2} = 1.8$ obtained by averaging experimental data for 64 DW pairs.

produces a texture with maximum light transmission at $\varphi_{\pi\pi} = \pi/4, 3\pi/4, 5\pi/4$, and $7\pi/4$ and extinction at $\varphi_{\pi\pi} = 0, \pi/2, \pi, 3\pi/2$, and $2\pi$, Fig. 6b, which is qualitatively similar to the experimental textures in Fig. 4.

To facilitate a comparison with the experiment, the width of the DW pairs is characterized by distances $L_{\pi/2}$ between the two central bright stripes, $L_\pi$ between two dark narrow stripes, $L_{3\pi/2}$ between two outermost stripes, Fig. 6b. $L_{\pi/2}$ measures the extension of mostly splay deformations between $\varphi_{\pi\pi} = 3\pi/4$ and $5\pi/4$, while the quantity $L_{3\pi/2} - L_{\pi/2}$ characterizes the extension of predominant bend. The characteristic width $\Delta x = \delta_{\pi\pi}\xi_{\pi\pi}$ appearing in Eq. (7) is close to $L_\pi$ when $\omega < 0.1$, but is smaller than $L_\pi$ when $\omega > 0.1$, as shown in Fig. 6d.

An increase of the elastic modulus $K$ makes the DWs wider and farther apart, to weaken the gradients of $\hat{\mathbf{n}}$ and $\mathbf{P}$. When the polar in-plane anchoring is weak, $\omega \ll 1$, the DWs are far away from each other, Fig. 6c, with $L_\pi = \Delta x \approx \sqrt{\frac{Kd}{2W_Q}}\ln\frac{4}{\omega}$ and a characteristic width $\xi_{\pi\pi} \approx \sqrt{\frac{Kd}{2W_Q}}\left(1 - \frac{\omega}{2}\right)$ close to $\xi$. The pair's energy approaches the sum of the energies of two individual $\pi$-solitons, $F_{\pi\pi} \approx 2F_\pi\left[1 + \frac{\omega}{2}\left(1 + \ln\frac{4}{\omega}\right)\right]$. A larger $\omega$ pushes the walls towards each other, shrinking the narrow $\mathbf{P}\downarrow\downarrow\mathbf{R}$ stripe, where the polarization is in the local minimum of the anchoring potential, Fig. 6c, d.

The soliton-antisoliton $\pi\bar{\pi}$ or $\bar{\pi}\pi$ pairs with alternating $\pi$-rotations of $\mathbf{P}$ satisfying Eq. (5) with the boundary conditions $\frac{\partial\varphi}{\partial x}(\pm\infty) = 0$, $\varphi(\pm\infty) = 0$ and corresponding to the S-pairs, are illustrated in Supplementary Figs. 10b and 11. Finally, solutions in which the boundary conditions are $\frac{\partial\varphi}{\partial x}(\pm\infty) = 0$, $\varphi(\pm\infty) = \pm\pi$ are also possible; they exhibit interesting spreading dynamics, as shown in Supplementary Fig. 12.

**Numerical solutions for planar pairs of domain walls at $K_1 \neq K_3$.** For $\kappa \equiv K_1/K_3 \neq 1$ and $\theta(x,z) = 0$, the free energy per unit area of an $N_F$ cell, after integration over the cell thickness, writes

$$f = \frac{K_3 d}{2}\left(\kappa\cos^2\varphi + \sin^2\varphi\right)\left(\frac{\partial\varphi}{\partial x}\right)^2 + W_Q\sin^2\varphi - 2W_P\left(\cos\varphi - 1\right),$$
(9)

The first integral of the Euler-Lagrange equation is

$$\xi_3^2\left(\kappa\cos^2\varphi + \sin^2\varphi\right)\left(\frac{\partial\varphi}{\partial x}\right)^2 - \sin^2\varphi + 2\omega\left(\cos\varphi - 1\right) = 0, \quad (10)$$

where $\xi_3 = \sqrt{\frac{K_3 d}{2W_Q}}$ is the extrapolation length associated with the bend modulus and quadrupolar anchoring. Equation (10) could

be solved numerically if rewritten as an expression describing a dynamic "particle" of a kinetic energy $\frac{1}{2}\xi_3^2\left(\frac{\partial\varphi}{\partial x}\right)^2$ (with the coordinate $x$ representing "time") rolling through a double-welled potential $V[\varphi] = \frac{2\omega(\cos\varphi - 1) - \sin^2\varphi}{2(\kappa\cos^2\varphi + \sin^2\varphi)}$, with zero total energy:

$$\frac{1}{2}\xi_3^2\left(\frac{\partial\varphi}{\partial x}\right)^2 + V[\varphi] = 0. \quad (11)$$

The $\pi\pi$-soliton solution corresponds to the particle rolling down the potential $V[\varphi]$ starting at $\varphi = 0$, where $V = 0$, through the two wells, and arriving at $\varphi = 2\pi$. Because energy is conserved, the soliton would be stable as the maxima at $\varphi = 0, 2\pi$ are both at $V = 0$. To find $\varphi(x)$, one needs to impart a small initial "momentum" forcing the particle to start the motion.

Figure 7 shows the results of numerical analysis. The width parameters $L_{\pi/2}, L_\pi$, and $L_{3\pi/2}$ of the DW pairs are not much affected by the elastic anisotropy when $K_1/K_3 \ll 1$, but increase, approximately as $L_\pi \propto \sqrt{K_1/K_3}$, when $K_1/K_3 > 1$, Fig. 7b. Because of their topological $2\pi$-rotation nature, the DW pairs must incorporate both splay and bend, no matter the value of $K_1/K_3$. A notable qualitative feature of the director profile $\varphi(x)$ of the DW pairs is that as $K_1/K_3$ increases, the stripes of splay widen, Fig. 7a. The structure tends to decrease the high splay energy by extending the length over which the splay develops; in contrast, it could afford a shorter bend development since $K_3$ is low. Domain walls in a chiral smectic C (SmC*) stabilized by a magnetic field show similar features[40], with the difference that, in SmC*, it is $K_3$ that is increased by the ionic screening. Thus, it is the bend stripes that are wider in SmC* than their splay counterparts.

The effect of elastic anisotropy on the ratio $L_{3\pi/2}/L_{\pi/2}$ is very strong when $K_1/K_3$ is in the range 0.1–10, Fig. 7c. As $K_1/K_3$ increases, the width of the splay region progressively expands and $L_{\pi/2}$ approaches $L_{3\pi/2}$. When compared to the experimental value $L_{3\pi/2}/L_{\pi/2} = 1.8$ obtained by averaging data of 64 DW pairs of both W and S types, the model of a planar $\pi\pi$-soliton suggests $K_1/K_3 \sim 10$ if $\omega = 0.1$. A more detailed comparison with the experiment is given below.

**Pairs of domain walls with polar tilts and $K_1 \neq K_3$.** The planar model neglects the possibility of director tilts towards the $z$-axis. Unless the cells are very thin, such a possibility should not be ignored. Figure 4e demonstrates that the director indeed tilts away from the $xy$-plane. To explore the effect, we return to Eq. (4) and use the ansatz in Eq. (3) for the tilt angle $\theta(x,z)$. For small $\theta$,

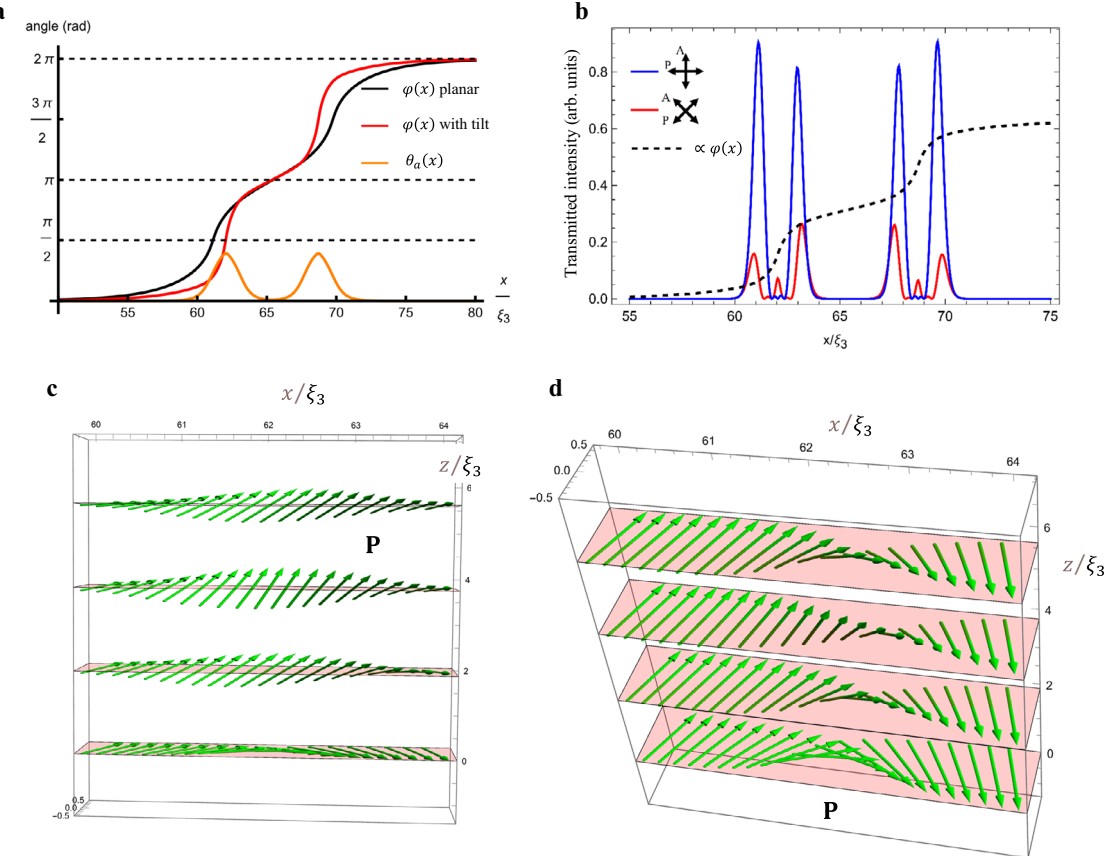

**Fig. 8 Simulated $\pi\pi$ soliton-soliton domain walls with non-zero polar tilt. a** Tilt magnitude $\theta_a(x)$ and polar angle $\varphi(x)$ profiles calculated by numerical minimization of the Frank–Oseen energy in Eq. (4) using the ansatz in Eq. (3) for $K_1/K_3 = 10$. The largest tilt occurs near $\varphi = \frac{\pi}{2}, \frac{3\pi}{2}$. **b** Transmitted light intensity through a cell and a filter of the type shown in Fig. 4e, where the wavelength $\lambda$ of light is chosen such that $\frac{\pi d \Delta n}{2\lambda} = \pi$. Note the favorable comparison between these results and the experimental data in Fig. 4c, f. Transmission is strong whenever $\varphi = \frac{\pi}{4}, \frac{3\pi}{4}, \frac{5\pi}{4}, \frac{7\pi}{4}$. **c, d** Two projected schemes of the polarization field in the one-quarter of the $\pi\pi$-soliton in which we find the largest tilt $\theta$, with the same parameters as in (**a**). In all simulations, $d = 15\,\xi_3$, $K_2 = K_3/2$, and $\omega = 0.1$.

the Frank–Oseen free energy density per unit area of the cell is

$$
\begin{aligned}
f = {} & \frac{dK_1}{2}\left[ \frac{2\pi^2\theta_a^2}{d^2} + \left(1 - \frac{\theta_a^2}{2}\right)\cos^2\varphi\,(\partial_x\varphi)^2 - \sin\varphi\cos\varphi\,\theta_a\partial_x\theta_a\partial_x\varphi \right] \\
& + \frac{dK_2}{4}\left(\cos\varphi\,\partial_x\theta_a + \sin\varphi\,\theta_a\partial_x\varphi\right)^2 \\
& + \frac{dK_3}{2}\left[ \left(1 - \theta_a^2\right)\sin^2\varphi\,(\partial_x\varphi)^2 + \sin^2\varphi\,\frac{(\partial_x\theta_a)^2}{2} \right] \\
& + W_Q\sin^2\varphi - 2W_P(\cos\varphi - 1).
\end{aligned}
\tag{12}
$$

Equation (12) demonstrates that in areas of strong splay, where $\cos^2\varphi\,(\partial_x\varphi)^2$ is large, a non-zero tilt $\theta_a > 0$ decreases the splay contribution by introducing twist (the terms proportional to $K_2$). The introduction of tilt becomes energetically costly when the cell is thin, with the tilt magnitude bounded by $\theta_a \lesssim \frac{d}{2\pi\xi_3\sqrt{2}} = \frac{1}{2\pi}\sqrt{\frac{W_Q d}{K_3}}$. For a 6.8 μm cell, $K_3/W_Q = 1$ μm, we expect $\theta_a \lesssim 0.4$. Significantly thinner cells would hardly experience polar tilt at all: a strong zenithal anchoring (associated with the tilts away from the $xy$ plane) makes the energetic costs of a vertical gradient over a short $d$ prohibitively high. Note, however, that our analysis is limited to a particularly simple $z$-dependence for both $\theta$ and $\varphi$ and the quantitative estimates above might be changed by a more rigorous analysis.

To find the tilt configuration $\theta_a(x)$, we minimize the Frank–Oseen free energy in Eq. (4) using gradient descent. The sharp bend of $\varphi(x)$ at large $K_1/K_3$ introduces computational challenges. To get a qualitative picture while ensuring the numerical convergence of the gradient descent procedure, we take $K_1/K_3 = 10$ and $d/\xi_3 = 15$, for which we expect a noticeable tilt. The resulting configurations of the polar angle $\varphi(x)$ and the tilt $\theta_a(x)$ are shown in Fig. 8a, c, d. Note that the director reorients to point nearly vertically ($\theta_a$ approaches $\pi/2$) in the middle of each of the two $\pi$-solitons, Fig. 8a. In these high tilt regions, the polar angle $\varphi$ rotates very rapidly as a function of $x$. This allows for a lower anchoring free energy as $\varphi$ maintains values close to $0, 2\pi$ for a larger range of $x$, Fig. 8a. The introduction of the tilt reduces the domain wall energy by about 20%, Fig. 9a. This measured decrease becomes even more substantial for larger values of $d/\xi_3$, Fig. 9a. It also represents a lower bound on the energy reduction as we use a constrained $z$-dependence of the azimuthal and polar angles. It would be interesting to minimize both $\varphi$ and $\theta$ without any constraints to find the true global energy minimum.

Taking the results for $\varphi, \theta$, we simulate the transmitted light intensities of the cell viewed through crossed polarizers with a monochromatic light of a particular wavelength $\lambda$, Fig. 8b. Choosing a wavelength at which the transmission through regions with $\mathbf{P}\!\uparrow\!\downarrow\!\mathbf{R}$ and $\mathbf{P}\!\downarrow\!\downarrow\!\mathbf{R}$ is suppressed, we find the results in Fig. 8b for two orientations of the polarizers. The simulated

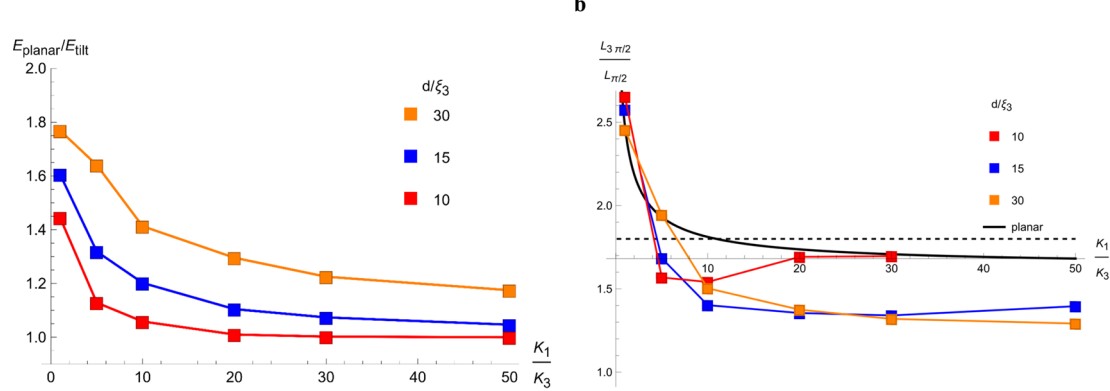

**Fig. 9 Characteristics of simulated DW pairs. a** Energy ratio of a planar domain wall ($\theta = 0$) versus one with a tilt ($\theta > 0$), as calculated from minimizing the Frank–Oseen energy in Eq. (4) using the ansatz in Eq. (3) for various values of $K_1/K_3$ and $d/\xi_3$. Note the marked energy gain from introducing a tilt for thick cells. For thinner cells, $d/\xi_3 < 10$, the gain is negligible, especially at large ratios $K_1/K_3$. **b** Ratio of width parameters $L_{3\pi/2}/L_{\pi/2}$ vs $K_1/K_3$ for different cell thicknesses $d/\xi_3$. Note that this ratio is expected to be smaller whenever there is substantial tilt in the director configuration. For thinner cells, $d/\xi_3 < 10$, the ratio approaches the planar value (black line) for large $K_1/K_3$ as the tilt becomes negligible. In all simulations, $\omega = 0.1$ and $K_2/K_3 = 0.5$. The dashed line shows $L_{3\pi/2}/L_{\pi/2} = 1.8$ obtained by averaging experimental data for 64 DW pairs. The lines connecting the data points in these plots are a guide to the eye.

intensities in Fig. 8b compare favorably to Fig. 4c (blue curve) and Fig. 4f (red curve). The red curve has an additional small peak between the two main peaks at around $\varphi \approx \frac{\pi}{4}, \frac{3\pi}{4}$. This small peak is not resolved in the experiment in Fig. 4f. One potential reason is that the intensity peak is very narrow, less than $\frac{\xi_3}{2}$; with $d = 6.8$ μm, $K_3/W_Q = 1$ μm, we expect $\frac{\xi_3}{2} \approx 0.9$ μm. Another reason is that the regions with $\theta > 0$ present a lower refractive index to the propagating beam as compared to the regions with $\theta = 0$; the index gradient bends the propagating rays away from the regions with $\theta > 0$ towards the regions with $\theta = 0$, which might further mitigate the small central peaks in the red curve in Fig. 8b. Note that the central peak would further narrow when the elastic anisotropy increases, $\frac{K_1}{K_3} > 10$, so that the reorientation of the angle $\varphi$ is even more rapid.

The tilted configurations depend sensitively on the cell thickness as $\theta_a$ decreases with decreasing $d/\xi_3$: the tilt becomes energetically less favorable since the director gradients along the $z$-axis become stronger under the condition of an infinite polar zenithal anchoring at the bounding plates. The ratio of the energy of a purely planar configuration, $E_{planar}$, to the energy of a configuration with a tilt, $E_{tilt}$, is shown in Fig. 9a. The numerical simulations suggest that the tilt is strongly reduced for $d/\xi_3 < 10$. For $K_3/W_Q = 1$ μm, and $d$ in the range (3–16) μm, one finds $2 < d/\xi_3 < 6$. Our experimental results are likely near the transition region when the tilt becomes energetically favorable, as suggested by the transmission peaks in Fig. 4e.

The width ratio $L_{3\pi/2}/L_{\pi/2}$ depends on the presence of tilt and the cell thickness, Fig. 9b. In thicker cells, the width ratio is smaller as the tilt allows for a faster reorientation of the azimuthal angle $\varphi$, as shown in Fig. 8a. The decrease, however, depends on the value of $K_1/K_3$, Fig. 9b. The dependence is subtle, with the width ratio approaching the planar value for small $K_1/K_3 \sim (1 - 4)$, but reaching a smaller value for $K_1/K_3 \approx 10$.

**Comparison of experimental and numerical shapes of the domain walls.** The width ratio $L_{3\pi/2}/L_{\pi/2}$ can be used to estimate $K_1/K_3$, Fig. 9b. We analyzed the profiles of transmitted monochromatic light intensities similar to the one in Fig. 4c for DWs pairs in samples of thickness ranging from 4.6 μm to 15.9 μm, which implies $3 < d/\xi_3 < 6$. In this range, there is no clear thickness dependence of the width ratio. The experimental data, averaged over 64 DWs pairs, yield $L_{3\pi/2}/L_{\pi/2} = 1.8 \pm 0.3$.

According to the model predictions in Fig. 9b, the value $L_{3\pi/2}/L_{\pi/2} = 1.8$ corresponds to $K_1/K_3 = (4 - 7)$ in the model with polar tilts and $d/\xi_3 = 10$, and to $K_1/K_3 = 10$ in the model of planar DWs. However, a relatively large standard deviation in the measured width parameter, $\pm 0.3$, embraces the possibility of much higher elastic anisotropy. An additional factor of uncertainty is in the strong dependence of the geometrical parameters and thus of $K_1/K_3$ on the in-plane polar anchoring parameter $\omega$, Fig. 7c. We thus conclude that the experiments on the structure of DW pairs place the lower bound on the elastic anisotropy of $N_F$, $K_1/K_3 \geq 4$, which is supported by both Figs. 7c and 9b.

## Discussion

The polar nature of the azimuthal surface anchoring of $N_F$ planar cells brings about patterns of polar monodomains and polydomains with alternating directions of the polarization **P**. Cooling the samples down to $N_F$ produces both **P↑↓R** and **P↓↓R** local surface alignments. These directions could be the same at the opposing plates, $\varphi(z = 0) = \varphi(z = d)$, or the opposite. In the latter case, the two different orientations must be connected by a twisted **P**, $\varphi(z) = (\varphi_d - \varphi_0)z/d + \varphi_0$, where $\varphi_0$ and $\varphi_d$ are the actual alignment directions at the two plates, which are found from the balance of the elastic and anchoring torques, Supplementary Eqs. (S5–S9). This twisted structure carries an energy $f_t = 2W_P + \frac{\pi^2}{2} \frac{K_2(1-\omega^2)}{d(1-\omega^2)+2\xi_2}$ per unit area, where $\xi_2 = K_2/W_Q$. In thin cells, $f_t$ could be large enough to eliminate the energy barrier between the $\varphi = 0$ and $\varphi = \pi$ states and cause the system to relax directly into the ground state $\varphi(z) = 0$, see Supplementary Eq. (S10) and Supplementary Fig. 13. In cells thicker than $d_c \approx \frac{\pi^2 K_2}{8W_P} \approx 3.6$ μm, $f_t$ is smaller than the energy $4W_P$ of the metastable uniform state $\varphi(z) = \pi$. In these thick cells, the local energy minimum at $\varphi = \pi$ and the energy barrier that separates $\varphi = \pi$ and $\varphi = 0$ directions are preserved (Supplementary Fig. 13); thus the system could relax into either the ground state, $\varphi(z) = 0$, or the metastable state $\varphi(z) = \pi$, which explains the observed domain structures with DWs in the thick samples.

We limited our analysis by the structures observed in the deep $N_F$ phase, but the experiments show rich dynamics of the emerging patterns during cooling in the high-temperature end of the $N_F$ phase, most likely caused by the temperature dependencies of $W_Q$, $W_P$, and the elastic constants; these will be described elsewhere.

A unique and unusual topological consequence of the surface polarity is that the DWs that separate domains of uniform polarizations form only as 360° pairs, of either the topologically protected soliton-soliton W-type or topologically trivial S-type. The DW pairs in which the order parameter varies from one global energy minimum to another while surpassing an energy barrier makes them similar to the DWs studied in cosmology models with "biased" vacuums[39], in which two vacuums have a slightly different energy and are separated by an energy barrier, similarly to the surface anchoring potential in Eq. (1) and Fig. 1g. In solid ferroics, surface interactions are polar along the normal to interfaces, which leads to the well-known patterns of alternating domains separated by 180° walls of the Bloch or Néel type[2,3]; 360° pairs could be observed only in the presence of an external field that competes with the apolar easy directions of the crystal structure[4]. DWs with 360° rotation of the director could also be observed in a smectic C liquid crystal[30,40–44], in which case they are attributed to an externally applied electric field[30] or to the asymmetry of the film along the normal direction[44]. In a uniaxial apolar nematic N, 360° DWs connect surface point-defects, called boojums, in a hybrid aligned film, Fig. 5a, in which one surface imposes a tangential orientation of $\hat{n}$ and another one sets a perpendicular alignment of $\hat{n}$, i.e., again the reason is the asymmetry with respect to the normal direction[38,45]. Under hybrid alignment of N, the 360° DW carries an elastic energy $\propto RL$ proportional to their length $R$ and width $L \ll R$, which is smaller than the elastic energy of an isolated boojum with an energy $\propto R^2$, where $R$ is the characteristic size of the system[45]. Unlike all listed examples, the 360° DWs in $N_F$ are caused by interactions that are polar in the plane of the bounding surfaces. The observed 360° pairs of DWs are also different from 180° DWs in $N_F$ cells with an antiparallel assembly of buffed plates that preset twist deformations[13,14,24]. The coupling between the surface polarity and the bulk structures allows us to estimate the polar contribution $\omega \equiv \frac{W_P}{W_Q} \sim 0.1$ to the in-plane anchoring of $\mathbf{P}$.

When the surfaces impose no restrictions on the in-plane orientation of $\mathbf{P}$, $N_F$ films feature the conic-sections textures, Fig. 5b, c, similar to focal conic domain textures in a smectic A. In a smectic A, the predominant director deformations are splay, signaling $K_1/K_3 \ll 1$, while in $N_F$, the prevailing curvatures are bend, Fig. 5b, c, suggesting $K_1/K_3 > 1$. The last condition makes the $N_F$ textures also similar to the textures of developable domains in columnar phases in which bend is the only allowed deformation of the director[27]. When the elastic constants show a strong disparity, liquid crystal textures often respond by introducing additional deformation modes (such as the effect of splay-canceling[46] or structural twist in the N droplets[47–49]). The DW pairs are no exception: the experiments, Fig. 4e, and numerical analysis, Figs. 8, 9, suggest that the in-plane splay-bend of $\mathbf{P}$ could be accompanied by out-of-plane tilts of $\mathbf{P}$, which introduce the twist of $\mathbf{P}$ and reduce the overall energy of the DWs. The analysis of the experimentally observed DW pairs suggests $K_1/K_3 > 4$.

The geometry of the domains and DW pairs is defined primarily by the balance of the polar and apolar terms in the surface potential, suggesting potential applications as sensors and solvents capable of spatial separation of polar inclusions. The advantage of $N_F$ is that the material is fluid and is thus easy to process in various confinements. Since the domains form in an optically transparent and birefringent $N_F$ fluid with a high susceptibility to low electric fields, other potential applications might be in electro-optics, electrically-controlled optical memory, and grating devices.

## Methods

**Sample preparation and characterization.** The aligning agent PI-2555 and its solvent T9039, both purchased from HD MicroSystems are combined in a 1:9 ratio.

Glass substrates with ITO electrodes are cleaned ultrasonically in distilled water and isopropyl alcohol, dried at 95 °C, cooled down to the room temperature and blown with nitrogen. An inert $N_2$ environment is maintained inside the spin coater. Spin coating with the solution of the aligning agent is performed according to the following scheme: 1 s @ 500 rpm → 30 s @ 1500 rpm →1 s @ 50 rpm. After the spin coating, the sample is baked at 95 °C for 5 min, followed by 60 minutes baking at 275 °C. The spin coating produced the PI-2555 alignment layer of thickness 50 nm.

The PI-2555 layer is buffed unidirectionally using a Rayon YA-19-R rubbing cloth (Yoshikawa Chemical Company, Ltd, Japan) of a thickness 1.8 mm and filament density 280/mm² to achieve a homogeneous planar alignment. An aluminum brick of a length 25.5 cm, width 10.4 cm, height 1.8 cm and weight 1.3 kg, covered with the rubbing cloth, imposes a pressure 490 Pa at a substrate and is moved ten times with the speed 5 cm/s over the substrate; the rubbing length is about 1 m. Unidirectional rubbing of a polyimide-coated substrates is known to align a nematic in a planar fashion, with a small pretilt of the director $\hat{n}$. For example, the director of a conventional nematic 5CB in contact with a buffed PI-2555 makes an angle $3° \pm 1°$ with the substrate; the tilt direction correlates with the direction $\mathbf{R}$ of buffing[50]. The pretilt in $N_F$ is expected to be smaller because of the surface polarization effect, as evidenced by the fact that the optical retardance of the uniform domains equals $\Delta nd$; however, the rubbing is still expected to produce nanoscale in-plane polarity because of the separation of oppositely charged moieties.

Two PI-2555-coated glass plates are assembled into cells in "parallel" geometry, with the two buffing directions $\mathbf{R}$ at the opposite plates being parallel to each other. One plate contains a pair of parallel transparent indium tin oxide (ITO) stripe electrodes separated along the $\mathbf{R}$-direction by a distance $l = 5$ mm in the studies of monodomains and 3 mm in the case of polydomains. A Siglent SDG1032X waveform generator and an amplifier (Krohn-Hite corporation) are used to apply an in-plane dc electric field $\mathbf{E} = E(0, \pm 1, 0)$. The observations are limited to an area 1 mm² at the center of the gap. Since the cell thickness $d$ is much smaller than $l$, the electric field in this region is predominantly horizontal and uniform.

The films with degenerate azimuthal surface anchoring are prepared by depositing a thin DIO film onto the surface of glycerin (Fisher Scientific, CAS No. 56-81-5 with assay percent range 99–100 %w/v and density 1.261 g/cm³ at 20 °C) in an open Petri dish. A piece of crystallized DIO is placed onto the surface of glycerin at room temperature, heated to 120 °C, and cooled down to the desired temperature with a rate of 5 °C/min. In the N, SmZ$_A$ and $N_F$ phases, DIO spreads over the surface and forms a film of a thickness defined by the deposited mass. For example, in Fig. 5a, a film of a thickness 5 μm resulted from a deposited 2.55 mg of the material.

The optical textures are recorded using a polarizing optical microscope Nikon Optiphot-2 with a QImaging camera and Olympus BX51 with an Amscope camera. PolScope MicroImager (Hinds Instruments) is used to map the director patterns and measure the optical retardance.

**Textural simulations.** To simulate the optical transmission through the cell, we employ the Jones matrix formalism. Assuming light propagation along the $z$-axis, the polarization in the $xy$-plane is described by a two-component vector, with $\binom{1}{0}$ polarization along $\hat{x}$ and $\binom{0}{1}$ along $\hat{y}$. The cell is represented as a $2 \times 2$ matrix consisting of a product of elements corresponding to thin slices of the uniaxial material. Given a tilt $\theta(x, z)$ and polar angle $\phi(x)$ of the optical axis, a thin slab $i$ of material of thickness $\Delta z$ will modify the electric field polarization at position $(x, z_i)$ according to a sequence of rotations and a phase retardance:

$$M_i(z_i) = \begin{pmatrix} \sin\phi(x) & \cos\phi(x) \\ -\cos\phi(x) & \sin\phi(x) \end{pmatrix} \begin{pmatrix} e^{-i2\pi\Delta z\sigma_{ez}(x,z_i)/\lambda} & 0 \\ 0 & e^{-i2\pi\Delta z\sigma_{0z}(x,z_i)/\lambda} \end{pmatrix} \begin{pmatrix} \sin\phi(x) & -\cos\phi(x) \\ \cos\phi(x) & \sin\phi(x) \end{pmatrix},$$

where $\lambda$ is the wavelength of the light, which we take to satisfy $\lambda = (n_e - n_0)d/2$. The dielectric eigenvalues are $\sigma_{0z} = n_0 = 1.5$ and

$$\sigma_{0z}(x, z_i) = \frac{n_0 n_e}{\sqrt{n_e^2[\sin\theta(x, z_i)]^2 + n_0^2[\cos\theta(x, z_i)]^2}},$$

where $n_e = 1.7$. The entire cell consists of $N$ slabs, such that $N\Delta z = d$. The full optical matrix for the cell is given by the product

$$M = \prod_{i=1}^{N} M_i(z_i) = \begin{pmatrix} M_{11} & M_{12} \\ M_{21} & M_{22} \end{pmatrix},$$

where we take the locations $z_i$ to be the midplanes of the thin slabs: $z_i = -d/2 + (i - 1/2)\Delta z$. We then choose a large enough $N$ such that our matrix converges. Note that the intensity for crossed polarizers can be easily extracted from the matrix elements $M_{ij}$. We have the following expressions for the intensities when the polarizers are aligned along the $x$ and $y$ axes and when they are at 45° to these axes, respectively:

$$I_+ = |M_{12}|^2 \quad \text{and} \quad I_\times = \frac{1}{4}|M_{11} + M_{21} - M_{12} - M_{22}|^2$$

**Reporting summary**. Further information on research design is available in the Nature Research Reporting Summary linked to this article.

## Data availability

All data that support the plots within this paper and other findings of this study are available from the corresponding author upon reasonable request.

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

## Acknowledgements

The authors thank N.A. Clark for illuminating discussions and for providing us with refs. [37,40]. This work was supported by NSF grant ECCS-2122399 (ODL).

## Author contributions

B.B. performed the experiments on polydomain structures and analyzed the data; M.R. performed the experiments on monocrystal states and analyzed the data; H.W. designed and performed the synthesis, purification, and chemical characterization of DIO; P.K. performed studies of films at the surface of glycerin and analyzed the data, K.T. assisted in cell preparation and experiments; S.P. assisted in the experimental set-ups; M.O.L. performed numerical analysis, O.D.L. conceived the project, analyzed the data, and wrote the manuscript with the inputs from all co-authors. All authors contributed to scientific discussions.

## Competing interests

The authors declare no competing interests.
