## [Peer Review File · Nature Communications]

Soliton walls paired by polar surface interactions in a ferroelectric nematic liquid crystalREVIEWER COMMENTS

Reviewer #1 (Remarks to the Author):

The work presented in this paper is outstanding from several points of view. First of all it deals with fundamental issues concerning the ferroelectric nematic phase that has been discovered recently. The behaviour of this phase is very different from that of its non-polarised variant which was studied for decades. In particular, in the classical geometry of a thin layer sandwiched between unidirectionally buffed plates, astonishingly rich variety of phenomena is observed. A sharp eye and clear mind are necessary to disentangle all subtle details. One of them is a very unusual type of the anchoring of this phase on unidirectionally buffed polymer surfaces. Another one concerns topologically stable 360° walls. The referee is convinced that this work will be appreciated not only by experts in liquid crystals but also by a large community of researchers in cosmology or condensed matter. In conclusion, the referee recommends publication of this paper.

Reviewer #2 (Remarks to the Author):

SUMMARY

This is a pioneering study of defect wall textures in a ferroelectric nematic liquid crystal and definitely deserves Nature Communication publication. I am happy with the observations and modeling with the following exception.

I believe that there needs to be included in this paper a discussion of the implications of these observations and their interpretation with respect to electrostatics. Spatial variation of the director/polarization orientation generates space charge density according to $\rho = \text{div}P$. These charges can be found in the bulk LC and on the surfaces. The authors mention Ref. 28 as a justification for ignoring such effects, but this inference is not obvious: In ref. 28 $P \sim \text{nanoC/cm}^2$, whereas in the NF $P \sim \text{microC/cm}^2$, making space charge effects 1000x stronger in the NF. This ratio appears in the polarization screening length $\xi_P = \text{SQRT}[\epsilon K/P^2]$. Estimating this in DIO without ions gives $\xi_P \sim$ a few nanometers. If this is the case then the space charge due to P would be the dominant effect. On p113 of the citation

[Zhiming Zhuang, Joseph E. Maclennan, and Noel A. Clark "Device Applications of Ferroelectric Liquid Crystals: Importance of Polarization Charge Interactions", Proc. SPIE 1080, Liquid Crystal Chemistry, Physics, and Applications, (25 July 1989); <https://doi.org/10.1117/12.976407>]

is a reference calculating the structure of 2π walls like those in this paper in the regime dominated by polarization space charge (no ions to screen). Calculated here is the 1 dimensional case which can be solved analytically. The result is that the width of the 2π wall is $\sim \xi_P$, controlled by the competition of Frank elasticity and space charge. The analysis in the current paper ignores the space charge interactions. For this to be correct, the effective P would have to be sufficiently small that ξ_P would be large compared to the length scales of the experiment, in this case microns, i.e. ξ_P would have to be a few tens of microns. So the ions are actually having a BIG effect here, screening out all but $\sim <.01\%$ of the polarization charge in the domain walls studied and leaving only frank elasticity and P^*E . terms

operative. This needs to be mentioned at least. Partial screening also leads to a contribution of the polarization charge to K splay

[Dynamics of the molecular orientation field coupled to ions in two-dimensional ferroelectric liquid crystals

Robert A. Pelcovits, Robert B. Meyer, and Jong-Bong Lee
PHYSICAL REVIEW E 76, 021704 (2007)].

COMMENTS

Every figure caption should give the cell thickness.

Reviewer #3 (Remarks to the Author):

Please see the review attachment.

Reviewer #4 (Remarks to the Author):

The recent discovery of the ferroelectric nematic phase is one of the most important in the field in the past decade or so (if not longer). This paper represents an early study of these fascinating and potentially ground-breaking materials. As such it is very suited to publication in Nature Communications, and will be of great interest not just to those in the field of liquid crystals, but also for wider physics community interested in magnetic materials, and topological systems. It is well written and clear throughout, and the results presented well. I have no hesitation in accepting it for publication. I have a number of suggested areas that the authors should consider, and minor corrections required, but feel that the thinking stimulated in me is merely an indication of the interest that the paper would induce. That is, the authors may wish to make changes based on some of the comments, but the paper would be acceptable without the comments being addressed.

Comments for consideration.

The model analyses domain walls for the polar nematic in terms of azimuthal and polar surface anchoring, showing that the domain walls invert the polarisation direction through change in azimuthal orientation of the director from n to $-n$. There are two limiting options for this inversion: either change in azimuth as described (in the xy plane), or through change of the zenithal (or tilt) angle (yz plane) of the n director. However, there is no discussion or mention of the latter option. For the thin samples, the formation of Bloch walls is justified if K_{22} is the lowest elastic constant. For the thicker samples, the polarising microscope study of the samples shown in figure 2 suggest that the domain is then a Neel wall. For a Neel wall, the rotation of n remains in the cell (xy) plane. Given K_{22} is the lowest elastic constant, the formation requires consideration of the zenithal anchoring term, in addition to the azimuthal and polar anchoring. This makes the situation more complicated. The fact that the results of figure 2a and d support the model of azimuthal change (ie Neel wall) suggests that the zenithal anchoring is far higher than the azimuthal anchoring energy. Some recognition of the factors that dictate the formation of either Bloch or Neel walls, such as the arguments suggested here. In addition to this, Figure

2 should include a photograph of the Neel walls with the crossed polarisers at an intermediate angle (eg 45°) to y. This would enable any change in retardation, and hence tilt and Bloch wall, to be emphasised. The waveplate results are a strong suggestion of a Neel wall type; the lack of retardation change would add proof. On Page 8; Paragraph 1, line 5: the elastic deformation for the Neel wall uses a mean elastic constant K. It should be made clear that this mean is of the splay and bend terms only, since the Neel wall (in 2D at least) does not include twist deformation.

The explanation of the anchoring on page 5 is a little unclear. Results for the required threshold fields when filled parallel and antiparallel to the rubbing direction are different. The use of a modulus sign for the former case but not the latter means that the reader must assume that E(down) changes from -1.0 to -1.4 kV/m. Also, the use of a single significant figure for the first measurement to two with the second means that the magnitude of the change (if any, since both are 1kV/m to one significant figure) is unclear. Moreover, the authors do not explain that these two sets of measurements were made after filling in the nematic (or NF) phase. The authors then state that the cells were subsequently studied after filling in this isotropic phase. At this point, it would be instructive to state what the new field thresholds are. How did this differ from filling perpendicular to the rubbing directions? Is there a difference in the domain structure / relative proportions of parallel and antiparallel to R that is dependent on filling direction?

The paper uses the proposed polar surface anchoring of a rubbed polyimide to explain the domain formation in the Ferroelectric Nematic. The relative sizes of domains formed parallel and antiparallel to the parallel rubbing directions is then related to the polar anchoring only. However, if there is a significant pretilt in the device, the parallel arrangement leads to a splay and bend elastic energy, whereas the anti-parallel does not. Presumably, the pretilt effect is considered to be negligible compared to the polar anchoring. The nematic pretilt is quoted in the methods section (which is read after the bulk of the text for this format) as 3° . The results of figure 1 suggest that there is not a difference in retardation between the two states, and hence the pretilt must remain small in the ferroelectric nematic phase.

Minor corrections:

Page 4 Paragraph 1, Lines 4,5: Polyimide 2555: More details are required such as supplier and nominal pretilt. Ideally, processing conditions such as thickness, rubbing strength and actual pretilt should be included, since the anchoring energies (including a new polar anchoring term) are determined.

Page 4 Paragraph 1, Lines 7,8: More details are required concerning the electric field and in particular, the electrode arrangement (top and bottom plates or just one plate; gap width etc). This is important since it is hard to apply an electric field without components in the z directions too.

Page 4, Paragraph 2, line 4: The symbol should be 1 - 2 μm , rather than using the division symbol. This is also true at line 9).

Page 4, Paragraph 2: I think the adjective "high" should refer to the liquid crystal sample, rather than that of the cell (since the glass substrates are irrelevant). Hence, cells with lower spacing, or thin samples" would be better (the latter because the sample is the material under study). This paragraph ought to include wavelength range studied for the

birefringence.

Figure 1 Caption. The wording of the caption for c) is slightly awkward. Perhaps" The polarisation P is anti-parallel to the rubbing direction R in the wider domains; and is parallel to R in the narrower domains"

Page 11, Paragraph 2, line 11: Spelling of monocrystal.

Figure 3: A Polarization Stabilized Kink. This structure is stable, maintaining a characteristic width $\xi_p = (K/P^2)^{1/2}$, in absence of applied electric field ($E_a = 0$). The stabilization is a result of the balance of Frank elasticity, tending to broaden the structure and the electrostatic attraction of the opposite signs of polarization charge, tending to narrow it.

2. POLARIZATION STABILIZED KINKS

In order to illustrate the general circumstances of nonuniform $\mathbf{P}(\mathbf{r})$ under which this field is significant we treat the following example. We consider a thick cell with an applied electric field \mathbf{E}_a along \mathbf{x} , which field has trapped a 2π reorientation of $\mathbf{P}(\mathbf{r})$ forming a disclination wall normal to the \mathbf{x} direction as shown in Figure 3. The sheets of polarization space charge and associated electric field are indicated. We now consider what happens when the applied electric field is reduced to zero. In more familiar case of a dielectric stabilized disclination π wall in a nematic liquid crystal, the wall has a characteristic width $w \sim \sqrt{K/\Delta\epsilon E_a^2}$, and spreads to infinite width as E_a goes to zero⁹. The length scale w is set by a balance of Frank elastic torque (K/w^2 , where K is the Frank elastic constant) and electric torque ($\Delta\epsilon E_a^2$, where $\Delta\epsilon$ is the dielectric anisotropy), and, as the applied field goes to zero, both the induced polarization and total electric field do also.

By contrast, when E_a is reduced to zero in the FLC case, the electric field $E_p = 4\pi P[1 - \cos(\phi)]$ of the polarization charge sheets remains unchanged in magnitude at the wall center ($x = 0$) and, as the wall spreads, the volume of space having comparable E_p increases, thereby increasing the electrostatic energy. The net wall energy per unit area, electrostatic (wP^2) plus elastic (K/w^2) is minimized for $w \sim \xi_p = \sqrt{K/P^2}$, so that the wall is stabilized by the polarization charge. We refer to this structure as a "Polarization Stabilized Kink (PSK)".

To treat this problem quantitatively we can write the torque balance equation⁹ for the variable ϕ , $K\nabla^2\phi = \tau_E$, where $\tau_E = PE\sin(\phi)$ is the field induced torque in $\mathbf{n}-\mathbf{P}$ field, using $E = E_a + E_p$ to give:

$$K \frac{d^2\phi}{dx^2} = (PE_a + 4\pi P^2)\sin(\phi) - 2\pi P^2 \sin(2\phi). \quad (1)$$

The second term, arising from the polarization space charge, remains when $E_a = 0$. Interestingly, it is of exactly the ϕ dependence that would obtain for the dielectric torque in a nonzero applied field⁹. Equation (1) is of the form $\phi_{ss} = \sin(\phi) + 2\eta\sin(2\phi)$ which may be solved¹⁰ for $\phi(x)$ to give the structure of Figure 3:

(a) for $0 > \eta > -\frac{1}{4}$,

$$\tan\left[\frac{\phi}{2}\right] = -(1 + 4\eta)^{-\frac{1}{2}} \operatorname{cosech}[(1 + 4\eta)^{\frac{1}{2}} s]; \quad (2)$$

(b) for $\eta = -\frac{1}{4}$,

$$\cot\left[\frac{\phi}{2}\right] = -s. \quad (3)$$

This paper deals with the domain walls (DWs) in the emerging N_F phase of DIO. The presentation starts with polarizing optical microscopy (POM) observations of N_F electric field response, giving rise to the connection between the directionalities of polarity vector \mathbf{P} and the electric field \mathbf{E} . Then, using synparallel buffered cells, the authors demonstrated a full switching of bulk polarity, driven by the switching of the surface polarity (? not very clearly presented), at sufficient electric field strengths. They showed an important observation that the thresholds of the switching electric field strength for $\mathbf{P} \uparrow \downarrow \mathbf{R}$ and $\mathbf{P} \downarrow \downarrow \mathbf{R}$ are distinct. They attributed this switching asymmetry to the mixing of quadrupolar and polar effects in the surface anchoring term. Finally, they discussed two types of DWs, which they claim as commonly seen, in the N_F phase. Overall, the paper is a great extension of the previous works, covering experimental and analytical considerations of surface anchoring and topology in detail, being certainly of broad interest. However, the manuscript, in its current form, suffers from some doubts for its core discussion (topology of W-pair of DW). This is serious enough to prevent the reviewer for supporting the publication. The problems and specific issues that should be solved for further consideration in NC are listed as below.

Critical issue

Figure 2 and related discussion:

The authors claim the corresponding director field looks like Fig. 2c. This is unsupported. First, the polscope is not rigorously helpful for studying DWs with a director field deformed (especially for twists) in cell normal direction. In such a case, the technique can just give some rough estimation. Second, without clarifying what the black lines observed in POM are, just extinction without defects or line disclination, the discussion made afterward in the manuscript is never justified. Indeed, in previous studies (Chen et al, Proc. Natl. Acad. Sci. U.S.A. 117, 14021 (2020); Li et al., Sci. Adv. 7, eabf5047 (2021)), similar black lines as line disclinations are reported and attributed to a Neel type DW. The reviewer supposes the previously reported topology is consistent with the S-pair of DW presented in the current manuscript. If they were lines, the pictures they have drawn would be totally wrong. The reviewer suggests them to check textures without polarizers and make confocal microscopy if possible. Based on these results, they may need updating the director field model and include all the relevant data. Furthermore, the presentation of 3D director field is necessary at least in the supplementary information, which helps readers to check whether the model is satisfactory to the POM observation. Of course, more discussions on the experimental finding and how they derive the reasonable director field are surely needed. Furthermore, there is a strong correspondence between the observations in the manuscript and those in the above Refs. The similarity and relationship for W- and S-pair of DWs to the prior works must be discussed.

Just extinction under crossed polarizers
or more like line disclination?

Major issue

1) The presentation in the manuscript lacks precision, needing considerable improvement. For example, page 4, “Ferroelectric monodomains in thin NF cells. ...” The presentation is contradictory to Fig. 1f. P vector aligns parallel to rubbing direction R there. However, the manuscript says P aligns antiparallel to R.

2) Page 5-7. The full switching of the bulk polarity (that can preserve for days even after switching off electric field) may (should?) arise as a result of surface polarity switching. Is it true or did I misunderstand something? It is helpful for them to give an explanation on the process microscopically: e.g. how the surface polarity switches with electric field? How the process links to the theoretical model suggested? Why the surface polarity after electric field off will not go back to initial anchoring? Does it strongly suggest the quadrupolar interaction is more dominant in as they discussed? As such, I suggest they make a more comprehensive discussion between the theoretical outcome and experimental results.

3) Page 6. The model only considers the simplest polar coupling with electric field. The flexoelectricity, electrostatic force, etc are neglected. For simple phenomenological description, the discussion is okay. However, they use the analytical solution for deducing the anchoring coefficients. They must give a more in-depth discussion, e.g. why each of the effects can be small enough (now only one was mentioned)? How much estimation error the exclusion of the effects will cause?

Minor issue

1) The reviewer found many typos, e.g. $d=1 \div 2 \mu\text{m}$ should read $d=1-2 \mu\text{m}$, DOI should read DIO, monocrystal should read monocrystal, etc.

2) The references are messy. For example, Refs. 5-20. Better to classify, and remove less-relevant and add more-relevant papers. Ref. 16 reports polar cholesterics with nothing linking to new synthesis and evaluation of the N_F phase presented here. Refs. 17 and 20 are nice pieces of pure theory and analysis but, to the reviewer’s option, do not fall into the category of new synthesis and evaluation of N_F phase. Instead, Manabe et al. synthesized a new N_F material with direct Iso- N_F phase transition (Manabe et al. *Liq. Cryst.* 48, 1079 (2021)). Li et al also reported tens of new N_F molecules (Li et al., *Sci. Adv.* 7, eabf5047 (2021); Li et al. *J. Am. Chem. Soc.* 42, 17857 (2021)), but all these are not cited. The authors should cite at least the most relevant papers, not as a matter of priority, but as one directed toward understanding the topic in a comprehensive manner. This is not all that related to the management of the references, so please check all.

- 3) Page 5. The difference in the electric fields... one global at $\varphi = 0$, and another local at $\varphi = \pm\pi$. Better to add the explanation of the experimental condition at the end of the sentence, e.g, "... for synparallel buffered cells", because the two-minima discussion is only valid for a rubbed cell case.
- 4) Cite the relevant papers for many places for comparison, especially for texture parts: e.g. page 6, "Cooling a cell with $d=4.7 \text{ um}$ from the SmZ_A phase results in a quasiperiodic domain texture of N_F , with ..."
- 5) The anchoring ratio ω and the elastic length ξ control the separation $\Delta x = \delta_{\pi\pi} \xi_{\pi\pi}$ between.... They must better define Δx , e.g. they can include the definition into figures.

Below we address all points raised by the reviewers.

Reviewer #1:

1. The work presented in this paper is outstanding from several points of view.

First of all it deals with fundamental issues concerning the ferroelectric nematic phase that has been discovered recently.

The behaviour of this phase is very different from that of its non-polarized variant which was studied for decades. In particular, in the classical geometry of a thin layer sandwiched between unidirectionally buffed plates, astonishingly rich variety of phenomena is observed. A sharp eye and clear mind are necessary to disentangle all subtle details.

One of them is a very unusual type of the anchoring of this phase on unidirectionally buffed polymer surfaces. Another one concerns topologically stable 360° walls.

The referee is convinced that this work will be appreciated not only by experts in liquid crystals but also by a large community of researchers in cosmology or condensed matter.

In conclusion, the referee recommends publication of this paper.

A1.1. We appreciate the assessment of our paper and the recommendation to publish.

Reviewer #2:

2.1. This is a pioneering study of defect wall textures in a ferroelectric nematic liquid crystal and definitely deserves Nature Communication publication.

A2.1. We appreciate the assessment of our paper and the recommendation to publish.

2.2. I am happy with the observations and modeling with the following exception. I believe that there needs to be included in this paper a discussion of the implications of these observations and their interpretation with respect to electrostatics. Spatial variation of the director/polarization orientation generates space charge density according to $\rho = \text{div}P$. These charges can be found in the bulk LC and on the surfaces. The authors mention Ref. 28 as a justification for ignoring such effects, but this inference is not obvious: In ref. 28 $P \sim \text{nanoC/cm}^2$, whereas in the NF $P \sim \text{microC/cm}^2$, making space charge effects 1000x stronger in the NF. This ratio appears in the polarization screening length $\lambda = \sqrt{\epsilon K/P}$. Estimating this in DIO without ions gives $\lambda \sim$ a few nanometers. If this is the case then the space charge due to P would be the dominant effect. On p113 of the citation [Zhiming Zhuang, Joseph E. MacLennan, and Noel A. Clark "Device Applications of Ferroelectric Liquid Crystals: Importance of Polarization Charge Interactions", Proc. SPIE 1080, Liquid Crystal Chemistry, Physics, and Applications, (25 July 1989); <https://doi.org/10.1117/12.976407>] is a reference calculating the structure of 2π walls like those in this paper in the regime dominated by polarization space charge (no ions to screen). Calculated here is the 1 dimensional case which can be solved analytically. The result is that the width of the 2π wall is $\sim \lambda$, controlled by the competition of Frank elasticity and space charge. The analysis in the current paper ignores the space charge interactions. For this to be correct, the effective P would have to be sufficiently small that λ would be large compared to the length scales of the experiment, in this case microns, i.e. λ would have to be a few tens of microns. So the ions are actually having a BIG effect here, screening out all but $\sim <.01\%$ of the polarization charge in the domain walls studied and leaving only frank elasticity and $P \cdot E$ terms operative. This needs to be mentioned at least. Partial screening also leads to a contribution of the polarization charge to K splay [Dynamics of the molecular orientation field coupled to ions in two-dimensional ferroelectric liquid crystals Robert A. Pelcovits, Robert B. Meyer, and Jong-Bong Lee PHYSICAL REVIEW E 76, 021704 2007].

A2.2. We agree with the Reviewer that the divergence-induced charges and their screening and associated renormalization of the splay elastic constant are all important factors. We stressed that the ionic impurities might screen the bound charges caused by $\text{div} P$. In the revised manuscript, we substantially expanded the discussion, by adding the consideration of the domain walls in the absence of the ions, stressing that the observed width is much larger than the one expected when the $\text{div} P$ charges are not screened. We also added (a) the discussion of the role of the elastic anisotropy since the screening of the polarization charge would effectively increase the splay elastic constant, (b) the textural observations in the azimuthally degenerate films of the material that demonstrate prevalence of bend, and (c) analysis of the domain structure in the presence of elastic anisotropy. Since these parts are new and clearly identified as such, we do not highlight them, in order to facilitate reading.

2.3. Every figure caption should give the cell thickness.

A2.3. We added this information.

Reviewer #3:

3.1. This paper deals with the domain walls (DWs) in the emerging NF phase of DIO. The presentation starts with polarizing optical microscopy (POM) observations of NF electric field response, giving rise to the connection between the directionalities of polarity vector P and the electric field E . Then, using synparallel buffered cells, the authors demonstrated a full switching of bulk polarity, driven by the switching of the surface polarity (? not very clearly presented), at sufficient electric field strengths. They showed an important observation that the thresholds of the switching electric field strength for $p \uparrow \downarrow$ and $p \downarrow \downarrow$ are distinct. They attributed this switching asymmetry to the mixing of quadrupolar and polar effects in the surface anchoring term. Finally, they discussed two types of DWs, which they claim as commonly seen, in the NF phase. Overall, the paper is a great extension of the previous works, covering experimental and analytical considerations of surface anchoring and topology in detail, being certainly of broad interest.

A3.1. We appreciate the assessment of our paper and the expressed assessment that the work covers experiment and analytical theory and that it is certainly of broad interest.

3.2. However, the manuscript, in its current form, suffers from some doubts for its core discussion (topology of W-pair of DW). This is serious enough to prevent the reviewer for supporting the publication. The problems and specific issues that should be solved for further consideration in NC are listed as below.

A.3.2. We modified Figures 2 and 3 and added new textures taken in monochromatic light, Figure 4, which explains the topology and geometry of the domain walls in greater detail.

3.3. Critical issue

Figure 2 and related discussion: The authors claim the corresponding director field looks like Fig. 2c. This is unsupported.

First, the polscope is not rigorously helpful for studying DWs with a director field deformed (especially for twists) in cell normal direction. In such a case, the technique can just give some rough estimation.

A3.3. We thank the Reviewer for the opportunity to extend the presentation. Although our initial intent was to limit the discussion to the 360° topological features of the domain walls, in the revised manuscript we added a new Figure 4 that supports 360° in-plane realignments and also reveals details such as the tilt of the polarization. The absence of twists in the narrow and wide domains is supported by the textures such as Figure 4 and by the measurements of the optical birefringence (now added to the Supplementary Material, Section II, and Supplementary Figure 8) and retardance. The PolScope technique is appropriate for the narrow and wide domains since the measured optical retardance is equal to the birefringence times the cell thickness; if there

would be a twist, the reading would be indeed different, we totally agree with the Reviewer. We used PolScope for the thin monodomain samples, in which the studies show the optical retardance to be equal $\Delta n d$ and that the samples become extinct whenever the rubbing direction is parallel or perpendicular to the polarization direction of one of the two crossed microscope's polarizer. In these monodomains, there is no twist.

3.4. Second, without clarifying what the black lines observed in POM are, just extinction without defects or line disclination, the discussion made afterward in the manuscript is never justified. Indeed, in previous studies (Chen et al, Proc. Natl. Acad. Sci. U.S.A. 117, 14021 (2020); Li et al., Sci. Adv. 7, eabf5047 (2021)), similar black lines as line disclinations are reported and attributed to a Neel type DW. The reviewer supposes the previously reported topology is consistent with the S-pair of DW presented in the current manuscript. If they were lines, the pictures they have drawn would be totally wrong. The reviewer suggests them to check textures without polarizers and make confocal microscopy if possible. Based on these results, they may need updating the director field model and include all the relevant data.

A3.4. The “lines” describe in previous publications are different from the 360° domain walls described in our manuscript. We added the discussion and comparison, following Fig.4:

The textures in Fig. 4a-c make it clear that the described DWs are indeed walls with a 360° reorientation of \mathbf{P} and \mathbf{i} , as opposed to the “bend texture with line disclination” of other NF materials presented by Li et. al.²¹, 180° surface disclination lines and 180° DWs described by Chen et. al.¹³ and Li et. al.²¹. The transmitted intensity profile in Fig.4c allows one to introduce the characteristic width parameters of the DW pairs: distances $L_{n/2}$ between the two central bright stripes, L_n between two dark narrow stripes, and $L_{3n/2}$ between two outermost stripes. These distances, although small, (8-15 μm), are clearly wider than the cores of singular disclinations, and 180° walls or surface disclinations described previously. Importantly, besides the in-plane 360° reorientation of \mathbf{P} and \mathbf{i} , the textures in $d = 6.8 \mu\text{m}$ cells also suggest tilts of these vectors away from the cell's xy plane, as described below.

We also added textures in azimuthally-degenerate films of the material, Fig.5; these show “Pure Polarization Reversal” defects similar to the ones reported in the cited pioneering work by Chen, Clark et al in 2020. We discuss the similarities and differences between these observations in the newly added section on the films.

3.5. Furthermore, the presentation of 3D director field is necessary at least in the supplementary information, which helps readers to check whether the model is satisfactory to the POM observation.

A3.5. We added a 3D model of the polarization field in Fig. 8c,d. This supplements the original 3D renderings such as the ones in Fig.2,3, which remain correct from the point of view of topology.

3.6. Of course, more discussions on the experimental finding and how they derive the reasonable director field are surely needed.

A3.6. We significantly expanded the analysis, adding the experimental data in Figures 4,5 and theoretical analysis in Figures 6-9.

3.7. Furthermore, there is a strong correspondence between the observations in the manuscript and those in the above Refs. The similarity and relationship for W- and S-pair of DWs to the prior works must be discussed.

A3.7. We expanded the description of the differences, which are substantial. The main feature of our domain walls is that the realignments are by 360° , as dictated by the unipolar anchoring with two minima of the anchoring potential. We added Fig. 4 and provided the explanation presented above as the answer A.3.4.

3.8. The presentation in the manuscript lacks precision, needing considerable improvement. For example, page 4, “Ferroelectric monodomains in thin NF cells. ...” The presentation is contradictory to Fig. 1f. P vector aligns parallel to rubbing direction R there. However, the manuscript says P aligns antiparallel to R.

A3.8. We are thankful to the Reviewer for noticing this misprint: the reference was supposed to be to part “1e”, not “1d”. We corrected the correspondence in the revised manuscript.

3.9. Page 5-7. The full switching of the bulk polarity (that can preserve for days even after switching off electric field) may (should?) arises as a result of surface polarity switching. Is it true or did I misunderstand something? It is helpful for them to give an explanation on the process microscopically: e.g. how the surface polarity switches with electric field? How the process links to the theoretical model suggested? Why the surface polarity after electric field off will not go back to initial anchoring? Does it strongly suggest the quadripolar interaction is more dominant in as they discussed? As such, I suggest they make a more comprehensive discussion between the theoretical outcome and experimental results.

A3.9. The polarity switches in the liquid crystal slab. The polarity of the substrate is not switched by the electric field. We added a statement to this effect on page 5:

Multiple cycles of switching show the same values of E_1 and E_2 , which means that the electric field realigns the polarization P in the liquid crystal bulk but does not switch the polarity of the rubbing direction R

3.10. Page 6. The model only considers the simplest polar coupling with electric field. The flexoelectricity, electrostatic force, etc are neglected. For simple phenomenological description, the discussion is okay. However, they use the analytical solution for deducing the anchoring coefficients. They must give a more in-depth discussion, e.g. why each of the effects can be small enough (now only one was mentioned)? How much estimation error the exclusion of the effects will cause?

A3.10. We added the discussion of the electrostatic and ionic effects. We also extended the analysis to the case of different elastic constants. We deduce the values of the polar contribution to the surface anchoring by realigning monocrystal samples, in which case the deformations do not carry flexoelectricity.

3.11. The reviewer found many typos, e.g. $d=1\div 2$ um should reads $d=1-2$ um, DOI should read DIO, monosrystal should read monocrystal, etc.

A3.11. We are thankful to the Reviewer for pointing out the misprints; we corrected these.

3.12. The references are messy. For example, Refs. 5-20. Better to classify, and remove less-relevant and add more-relevant papers. Ref. 16 reports polar cholesterics with nothing linking to new synthesis and evaluation of the NF phase presented here. Refs. 17 and 20 are nice pieces of pure theory and analysis but, to the reviewer's option, do not fall into the category of new synthesis and evaluation of NF phase. Instead, Manabe et al. synthesized a new NF material with direct Iso-NF phase transition (Manabe et al. *Liq. Cryst.* 48, 1079 (2021)). Li et al also reported tens of new NF molecules (Li et al., *Sci. Adv.* 7, eabf5047 (2021); Li et al. *J. Am. Chem. Soc.* 42, 17857 (2021)), but all these are not cited. The authors should cite at least the most relevant papers, not as a matter of priority, but as one directed toward understanding the topic in a comprehensive manner. This is not all that related to the management of the references, so please check all.

A3.12. We agree and added the references suggested by the Reviewer. We think that theoretical papers fall within the category of "evaluation of the NF phase", thus we keep these.

3.13. Page 5. The difference in the electric fields... one global at $\phi = 0$, and another local at $\phi = \pm \dots$. Better to add the explanation of the experimental condition at the end of the sentence, e.g, "... for synparallel buffered cells", because the two-minima discussion is only valid for a rubbed cell case.

A3.13. We added a clarification "in the cells with the parallel assembly of the buffed plates" as suggested.

14. Cite the relevant papers for many places for comparison, especially for texture parts: e.g. page 6, "Cooling a cell with $d=4.7$ um from the SmZA phase results in a quasiperiodic domain texture of NF, with ..."

A3.14. These are our original observations never reported before; thus we could not cite prior works while presenting the textures. We do cite Ref.22, which describes the studies of the DIO material. We also compare our defect structures to those presented in the earlier works whenever possible; particular examples are stated above. We also extended the comparison of SmZA to the splay N phase considered by Mertelj et al.

15. The anchoring ratio α and the elastic length l_e control the separation $\Delta = \dots$ between.... They must better define Δ , e.g. they can include the definition into figures.

A3.15. We agree and added the value of Δ in the plot in Fig.6d and in the statement that follows

the figure: “The characteristic width δ appearing in Eq.(7) is close to δ_0 when $\delta \ll \delta_0$ ”

0.1, but is smaller than . when $\tilde{n} > 0.1$,
Fig.6d.”

Reviewer #4:

4.1. The recent discovery of the ferroelectric nematic phase is one of the most important in the field in the past decade or so (if not longer). This paper represents an early study of these fascinating and potentially ground-breaking materials. As such it is very suited to publication in Nature Communications, and will be of great interest not just to those in the field of liquid crystals, but also for wider physics community interested in magnetic materials, and topological systems. It is well written and clear throughout, and the results presented well. I have no hesitation in accepting it for publication. I have a number of suggested areas that the authors should consider, and minor corrections required, but feel that the thinking stimulated in me is merely an indication of the interest that the paper would induce. That is, the authors may wish to make changes based on some of the comments, but the paper would be acceptable without the comments being addressed.

A.4.1. We thank the Reviewer for the evaluation of the paper and for recommending acceptance. We still decided to address all the comments from all reviewers, in order to improve the presentation.

4.2. Comments for consideration. The model analyses domain walls for the polar nematic in terms of azimuthal and polar surface anchoring, showing that the domain walls invert the polarisation direction through change in azimuthal orientation of the director from n to $-n$. There are two limiting options for this inversion: either change in azimuth as described (in the xy plane), or through change of the zenithal (or tilt) angle (yz plane) of the n director. However, there is no discussion or mention of the latter option. For the thin samples, the formation of Bloch walls is justified if K_{22} is the lowest elastic constant. For the thicker samples, the polarising microscope study of the samples shown in figure 2 suggest that the domain is then a Neel wall. For a Neel wall, the rotation of n remains in the cell (xy) plane. Given K_{22} is the lowest elastic constant, the formation requires consideration of the zenithal anchoring term, in addition to the azimuthal and polar anchoring. This makes the situation more complicated. The fact that the results of figure 2a and d support the model of azimuthal change (ie Neel wall) suggests that the zenithal anchoring is far higher than the azimuthal anchoring energy. Some recognition of the factors that dictate the formation of either Bloch or Neel walls, such as the arguments suggested here.

A.4.2. We agree and added an extended analysis of textures in Fig.4 and in the theory. We also stressed that the out-of-plane anchoring is much stronger than the azimuthal one, adding “Therefore, we expect that the out-of-plane (zenithal) polar anchoring is much stronger than the in-plane azimuthal anchoring” on page 5 and explicitly stating in the theoretical part on page 17:” Since the polar tilt at the bounding plates is penalized by a large surface charge, we assume that the zenithal polar anchoring is infinitely strong.” Moreover, our added theoretical analysis suggests that even in this case, a reorientation of the director along the normal to the cell is still possible.

4.3. In addition to this, Figure 2 should include a photograph of the Neel walls with the crossed polarisers at an intermediate angle (eg 45°) to y. This would enable any change in retardation, and hence tilt and Bloch wall, to be emphasised. The waveplate results are a strong suggestion of a Neel wall type; the lack of retardation change would add proof.

A.4.3. We agree and add a new Figure 4.

4.4. On Page 8; Paragraph 1, line 5: the elastic deformation for the Neel wall uses a mean elastic constant K. It should be made clear that this mean is of the splay and bend terms only, since the Neel wall (in 2D at least) does not include twist deformation.

A.4.4. We agree and added a clarification at the bottom of page 17: "However, analytical solutions useful for the understanding of the DW pairs could be found if $\delta(x, z) = 0$ and $K_1 = K_3 = K$; the planar geometry with $\delta = 0$ excludes twists."

4.5. The explanation of the anchoring on page 5 is a little unclear.

A.4.5. We added a clarification "in the cells with the parallel assembly of the buffed plates "

4.6. Results for the required threshold fields when filled parallel and antiparallel to the rubbing direction are different. The use of a modulus sign for the former case but not the latter means that the reader must assume that E(down) changes from -1.0 to -1.4 kV/m.

A.4.6. We try to use consistent notations in the revised manuscript.

4.7. Also, the use of a single significant figure for the first measurement to two with the second means that the magnitude of the change (if any, since both are 1kV/m to one significant figure) is unclear.

A.4.7. Thank you, we corrected the oversight.

4.8. Moreover, the authors do not explain that these two sets of measurements were made after filling in the nematic (or NF) phase.

A.4.8. Thank you, we added the conditions.

4.9. The authors then state that the cells were subsequently studied after filling in this isotropic phase. At this point, it would be instructive to state what the new field thresholds are. How did this differ from filling perpendicular to the rubbing directions? Is there a difference in the domain structure / relative proportions of parallel and antiparallel to R that is dependent on filling direction?

A4.9. Thank you, we added the information on page 7: “we avoid it by filling the cells in at 180°C and then rapidly cooling the sample through the N_F phase with a rate 30°C/min, followed by slow cooling through SmZA and N_F with the rate 2°C/min. Thin d=1.1 μm monodomain samples show E_L = -0.4 kV/m, E_T = 0.3 kV/m, which yields c₀ ≈ 0.07. With the values of P and K₂ above, one estimates W_Q ≈ 8.8 × 10⁻⁶ J/m², and W_P ≈ 0.63 × 10⁻⁶ J/m². In what follows, we discuss the data for cells filled in the isotropic phase at 180°C; the domain structures are similar to those

4.10. The paper uses the proposed polar surface anchoring of a rubbed polyimide to explain the domain formation in the Ferroelectric Nematic. The relative sizes of domains formed parallel and antiparallel to the parallel rubbing directions is then related to the polar anchoring only. However, if there is a significant pretilt in the device, the parallel arrangement leads to a splay and bend elastic energy, whereas the anti-parallel does not. Presumably, the pretilt effect is considered to be negligible compared to the polar anchoring. The nematic pretilt is quoted in the methods section (which is read after the bulk of the text for this format) as 3°. The results of figure 1 suggest that there is not a difference in retardation between the two states, and hence the pretilt must remain small in the ferroelectric nematic phase.

A4.10. We agree and added the following statement in the Methods:” The pretilt in N_F is expected to be smaller because of the surface polarization effect, as evidenced by the fact that the optical retardance of the uniform domains equals $\Delta n d$; however, the rubbing is still expected to produce nanoscale in-plane polarity because of the separation of oppositely charged moieties. ”

4.11. Minor corrections: Page 4 Paragraph 1, Lines 4,5: Polyimide 2555: More details are required such as supplier and nominal pretilt. Ideally, processing conditions such as thickness, rubbing strength and actual pretilt should be included, since the anchoring energies (including a new polar anchoring term) are determined.

A4.11. We expanded the description, by adding “The pretilt in N_F is expected to be smaller because of the surface polarization effect, as evidenced by the fact that the optical retardance of the uniform domains equals $\Delta n d$; however, the rubbing is still expected to produce nanoscale in-plane polarity because of the separation of oppositely charged moieties. ”

4.12. Page 4 Paragraph 1, Lines 7,8: More details are required concerning the electric field and in particular, the electrode arrangement (top and bottom plates or just one plate; gap width etc). This is important since it is hard to apply an electric field without components in the z directions too.

A4.12. We added a clarification:” The observations are limited to an area 1 mm² at the center of the gap. Since the cell thickness d is much smaller than 1, the electric field in this region is predominantly horizontal and uniform.”

4.13. Page 4, Paragraph 2, line 4: The symbol should be 1 - 2 μm, rather than using the division

symbol. This is also true at line 9).

A4.13. Thank you, we corrected the symbols.

4.14. Page 4, Paragraph 2: I think the adjective "high" should refer to the liquid crystal sample, rather than that of the cell (since the glass substrates are irrelevant). Hence, cells with lower spacing, or thin samples" would be better (the latter because the sample is the material under study). This paragraph ought to include wavelength range studied for the birefringence. **A4.14.** We included the data on wavelengths and added the plot of birefringence in the Supplementary material.

4.15. Figure 1 Caption. The wording of the caption for c) is slightly awkward. Perhaps" The polarisation P is anti-parallel to the rubbing direction R in the wider domains; and is parallel to R in the narrower domains"

A4.15. Thank you, we reworded as suggested.

4.16. Page 11, Paragraph 2, line 11: Spelling of monocrystal.

A4.16. Thank you, we tried to correct all the misprints.

We hope that the revised manuscript addresses the comments raised by the Reviewers and that it can be now accepted for publication.

Sincerely yours,

Oleg D. Lavrentovich,

REVIEWERS' COMMENTS

Reviewer #3 (Remarks to the Author):

The reviewer's concerns have been solved.

Reviewer #4 (Remarks to the Author):

I believe the referees have done a particular thorough analysis of this paper. I had thought the original work would be acceptable but that a number of changes would help improve the work. These changes and the ones induced by the other referees, certainly succeed in that and the paper is now very thorough indeed. I have no hesitation with accepting publication and believe that this will become a major work in the field.

With respect to the highlighted passages, I do have one comment that the authors should consider relating to the new text on page 4. At the risk of identifying myself, I think the paper repeats a common error in the field of liquid crystals that contradicts the terminology used in device engineering. When designing a liquid crystal cell for use in some application, the thickness is one of the dimensions that is specified. It is usually about 2 - 3 mm or so, and is dominated by the thickness of the glass and any films mounted on it. Personally, I think that "liquid crystal thickness", "cell spacing" or even "sample thickness" are preferable. Though those specialising in the field of liquid crystal science will think this slightly pedantic, the journal is aimed at a wider audience for whom that might confuse. This certainly has no bearing on whether the paper is accepted or not.

We are thankful to you and the Reviewers for the assessment of our manuscript. We followed the suggestion by **Reviewer 4**:

Reviewer 4 states the following:

I have no hesitation with accepting publication and believe that this will become a major work in the field.....

With respect to the highlighted passages, I do have one comment that the authors should consider relating to the new text on page 4. At the risk of identifying myself, I think the paper repeats a common error in the field of liquid crystals that contradicts the terminology used in device engineering. When designing a liquid crystal cell for use in some application, the thickness is one of the dimensions that is specified. It is usually about 2 - 3 mm or so, and is dominated by the thickness of the glass and any films mounted on it. Personally, I think that "liquid crystal thickness", "cell spacing" or even "sample thickness" are preferable. Though those specialising in the field of liquid crystal science will think this slightly pedantic, the journal is aimed at a wider audience for whom that might confuse. This certainly has no bearing on whether the paper is accepted or not.

Authors response:

Following the suggestion, on page 4, we changed “depending on the cell thickness d ” to “depending on the thickness d of the liquid crystal layer”

We also made sure that the panels in figures are labelled in the left upper corner and changed “a. u.” to “arb. units” in Figs.4,8.

=====

We hope that manuscript can be now sent to production.

Sincerely yours,

Oleg D. Lavrentovich,
